# Introducing FRIDA v2.1: A feedback-based, fully coupled, global integrated assessment model of climate and humans

William Schoenberg<sup>1,2</sup>, Benjamin Blanz<sup>3</sup>, Jefferson K. Rajah<sup>1</sup>, Benjamino Callegari<sup>4</sup>, Christopher Wells<sup>5</sup>, Jannes Breier<sup>6</sup>, Martin B. Grimeland<sup>4</sup>, Andreas Nicolaidis Lindqvist<sup>7,8</sup>, Lennart Ramme<sup>9</sup>, Chris Smith<sup>10,11</sup>, Chao Li<sup>9</sup>, Sarah Mashhadi<sup>12</sup>, Adakudlu Muralidhar<sup>12</sup>, and Cecilie Mauritzen<sup>12</sup>

Correspondence to: William Schoenberg (bschoenberg@iseesystems.com)

Abstract. The current crop of models assessed by the Intergovernmental Panel on Climate Change (IPCC) to produce their assessment reports lack endogenous process-based representations of climate-driven changes to human activities, especially beyond the purely economic consequences of climate change. These climate-driven changes in human activities are critical to understanding the co-evolution of the climate and human systems. Earth System Models (ESMs) that represent the climate system and Integrated Assessment Models (IAMs) that represent the human system are typically separate, with assumptions that create coherency coordinated through RCPs and SSPs in ScenarioMIP, the core scenario analysis protocol. This divide limits understanding of climate-human feedback. An alternative aggregated approach, which couples human and natural systems (CHANS) such as the one used to build the Feedback-based knowledge Repository for IntegrateD Assessments "FRIDA" v2.1 IAM documented here, integrates climate and human systems into a unified global model, prioritizing feedback dynamics while maintaining interpretability. FRIDA represents the Earth's radiation balance, carbon cycle, and relevant portions of the water cycle alongside human demographics, economics, agriculture, and human energy use. Built using the System Dynamics method, it contains seven interconnected modules. Each subsystem is calibrated to data and validated to ensure structurally appropriate behaviour representation. FRIDA demonstrates that an aggregate, feedback-driven modelling approach, capturing CHANS interconnections with rigorous measurements of uncertainty, is possible. It complements conventional IAMs by highlighting missing feedback structures that affect future projections. Our work with

of uncertainty, is possible. It Deleted: climate-human

Deleted:

Deleted: It

FRIDA suggests SSP1-Baseline, SSP2-Baseline, and SSP5-Baseline are all overly optimistic on the prospects for future

<sup>&</sup>lt;sup>1</sup>System Dynamics Group, University of Bergen, P.O. Box 7802, 5020 Bergen, Norway

<sup>&</sup>lt;sup>2</sup>isee systems inc., 24 Hanover St. Suite 8A, Lebanon, New Hampshire 03766, USA

<sup>&</sup>lt;sup>3</sup>Research Unit Sustainability and Climate Risks, University of Hamburg, Grindelberg 5, 20144 Hamburg, Germany

<sup>&</sup>lt;sup>4</sup>School of Economics, Innovation and Technology, Kristiania University of Applied Sciences, Oslo, Norway

<sup>&</sup>lt;sup>5</sup>School of Earth and Environment, University of Leeds, Leeds, LS2 9JT, United Kingdom

<sup>&</sup>lt;sup>6</sup>Potsdam Institute for Climate Impact Research, Telegrafenberg A31, 14473 Potsdam, Germany

<sup>&</sup>lt;sup>7</sup>Stockholm Resilience Centre, Stockholm University, Albanovägen 28, SE-106 91 Stockholm

<sup>&</sup>lt;sup>8</sup>RISE Research Institutes of Sweden, Ideon Beta5, Scheelevägen 17, 22370, Lund, Sweden.

<sup>&</sup>lt;sup>9</sup>Max-Planck-Institute for Meteorology, Bundesstraße 53, 20146 Hamburg, Germany

<sup>&</sup>lt;sup>10</sup>Department of Water and Climate, Vrije Universiteit Brussel, 1050 Brussels, Belgium

<sup>&</sup>lt;sup>11</sup>Energy, Climate and Environment Program, International Institute for Applied Systems Analysis (IIASA), Laxenburg, Austria

<sup>&</sup>lt;sup>12</sup>Department of Ocean and Ice, Norwegian Meteorological Institute, 0313 Blindern, Oslo

economic growth due to these feedbacks, while SSP3-Baseline and SSP4-Baseline, the SSPs with the highest challenges to

adaptation, align more closely with our results. Future work will further refine climate impact representations, energy
modelling, policy scenario creation, and stakeholder engagement for informed policymaking.

### 1 Introduction

60

Properly representing the co-evolution of the climate system with the humans who exist within it requires models that two-way couple climate processes with human processes that include but extend beyond conomic dimensions (Calvin and Bond-Lamberty, 2018; Donges et al., 2017; Motesharrei et al., 2016). This class of models is referred to as CHANS models, which stands for coupled human and natural systems models (Alberti et al., 2011; Kramer et al., 2017; Liu et al., 2007). The current crop of models assessed by the Intergovernmental Panel on Climate Change (IPCC) to produce their assessment reports lack the CHANS perspective of endogenous process-based climate-driven changes to human activities (Beckage et al., 2022; Donges et al., 2021; Wilson et al., 2021). These state-of-the-art models for understanding and assessing the impacts of global climate change are divided into two (increasingly overlapping) modelling efforts. First, there are Earth System Models (ESMs) that focus on modelling climate processes related to changes in the atmosphere, oceans, land, ice and biosphere. Second, there are Integrated Assessment Models (IAMs) that focus on modelling the human processes responsible for creating the aforementioned changes represented in ESMs. Global scale ESMs and IAMs are generally not directly coupled; instead, information from one is fed into the other, whether directly via exogenous inputs (as is sometimes done in ESMs to represent the future development of the human system), or indirectly via emulation (as is often done in IAMs to represent the response of the climate system to potential future human behaviour). In the IPCC's fifth and sixth assessment reports (AR5 and AR6), the assumptions governing this information exchange between the two classes of models were

The RCPs and SSPs were designed to ensure coherence in the scenarios generated by both ESMs and IAMs when either class of model individually lacked the structure necessary to represent known feedback which links humans to climate, and climate to humans. Any climate damage representations available in the IAMs used to generate the SSPs were explicitly switched off, to prevent double-counting of climate damages when the ESM output was fed into impact models, and also due to the lack of confidence in the systematic nature of these representations (O'Neill et al., 2014, 2020). This sequential approach (from future emissions to climate response, and then to resultant climate impacts) has been proposed to continue to be followed in the next round of ScenarioMIP for CMIP7 (Van Vuuren et al., 2025). While this is justifiable in the context of the overall framework, with different types of models focused heavily on different aspects of the human-climate system, it forecloses on the exploration of the impact of climate damages on the overall trajectory of the coupled system, limiting the plausibility of such scenarios (O'Neill et al., 2014). This is of particular importance in high-emissions scenarios, in which substantial climate damages should in reality be expected to alter the overall trajectory of the system (Van Vuuren et al.,

coordinated by the Representative Concentration Pathways (RCPs) (van Vuuren et al., 2011) and the Shared Socioeconomic

Pathways (SSPs) (Kriegler et al., 2012; O'Neill et al., 2014) respectively.

Deleted: the
Deleted: both economic and not
Deleted:
Deleted:
Deleted:

Deleted: what is known to be the true

Deleted: is

2025; Woodard et al., 2019). In essence, these scenario frameworks require each class of model to focus on one part of the overall emissions-climate-damages chain (Calvin and Bond-Lamberty, 2018; Donges et al., 2017). For the class of IAMs that do contain process-based climate representations either via the inclusion of reduced-complexity climate emulators or by directly hard-coupling to ESMs, the RCPs and SSPs are used to standardize, assumptions across scenarios and comparisons to other models, in effect potentially perpetuating the inconsistencies from the uncoupled models to the coupled models.

The disconnect in the modelling process has contributed to a division of responsibility for representing the co-evolution of climate and humans, making CHANS modelling a new frontier for global integrated assessment modelling (Li et al., 2023). That disconnect can be seen in both detailed process-based IAMs and cost-benefit IAMs (for terminology see Weyant (2017)) as an under representation of the impact of climate on human systems beyond direct or highly aggregated impacts on economic output. Although this schism is gradually becoming less pervasive throughout the climate change modelling community, it remains evident in the IPCC working groups (WGs): WG I focuses on the physical science basis of climate change; WG II on the impacts of, adaptation to, and vulnerability of life on earth, to climate change; and WG III on the mitigation of climate change. This organizational structure within the IPCC places the Earth system modelers generally in WG I, and the integrated assessment modelers are then spread between WG II and WG III depending upon their research aims (impact vs. adaptation or mitigation). These two modelling silos are quite helpful for tackling the practical realities of modelling climate, and its impacts at ever increasing levels of specificity (disaggregation), whether that be spatial, sectoral, or otherwise. The trade-off for this increased level of specificity in the ESMs and IAMs is a lack of explicit modelling of the grand system-wide feedback processes. Capturing these processes requires a CHANS perspective for a more complete representation of climate impacts beyond direct economic damages. Such a representation fully couples the climate and the

human world together – creating the feedback which locks these two subsystems into a synchronous co-evolution. In turn, this begs the question: is the increased specificity of these ever more disaggregated models, both ESM and IAM alike, coming at the expense of structural errors and scenario inconsistencies being introduced from the lack of an explicit, complete two-way coupling between climate and humans? Global models which fall into the CHANS category including ANEMI3 (Breach and Simonovic, 2021), En-ROADS (Kapmeier et al., 2021), E3SM-GCAM (Di Vittorio et al., 2025), and Felix (Eker et al., 2019; Ye et al., 2024) ultimately pose the same question although generally at higher levels of disaggregation, and therefore with challenges for interpretability.

105

Fully answering this question requires a model developed using an alternative, aggregated CHANS driven approach that is complementary to the current ESMs and IAMs as well as the more disaggregated CHANS models. First and foremost, the division between climate processes and human processes across models must be bridged. In the language of Donges et al. (2021), such an approach must represent the Biophysical taxon (j.e., the "natural laws" of physics, chemistry or ecology), Socio-cultural taxon (j.e., human behaviour and decision making), and the Socio-metabolic taxon (j.e., material interactions of the biophysical and human systems), with the necessary complexity and detail to capture the unique contributions of each set of interconnected dynamics to the evolution of the entire world-Earth system. This suggests that the two halves of the single, unified world-Earth system must be represented as equal partners in the same system of equations, eschewing the use of the

Deleted: se

Deleted: growing

Deleted:

Deleted: and IAM

Deleted: an

Deleted: still useful for

Deleted: ing

Deleted: (Di Vittorio et al., 2025)

Deleted: led

Deleted: ed

Deleted: cd

Deleted: or

Deleted: that

Deleted: A

Deleted:
Deleted:
Deleted: ,
Deleted: approach
Deleted: which include aspects of both already
Deleted: e.g
Deleted: .
Deleted: e.g
Deleted: e.g
Deleted: , and

Deleted: the Biophysical taxon

quantitative results from the SSPs to coordinate assumptions between them. Instead, as some of the more disaggregated CHANS models do, the thinking and narratives contained within the SSPs should be used to describe the potential for the future unfolding of human behaviour (see e.g., Eker et al., 2019; Ye et al., 2024). An aggregated CHANS driven modelling 140 approach that answers our question allows the SSPs to serve to categorize uncertainty in the presentation of future scenarios. Second, an aggregated CHANS driven modelling approach requires a fully endogenous, process-based explanation for model behaviour. Without it, an aggregated CHANS driven approach would not adequately address the main problem caused by current schism. Third, to increase the understanding derived from models built using an aggregated CHANSdriven approach, highly aggregated models are preferred (Robertson, 2021). Without these considerations, a model built following a more disaggregated CHANS approach risks building an opaque model which is far too complex to yield (actionable or trustworthy) insight. Finally, to avoid problems with the potential for the lack of precision that are inherent with a model constructed in this manner, the level of precision of a model built using the proposed approach must be carefully tracked, quantified, and presented via a systematic uncertainty analysis.

In this paper, we formally introduce the "Feedback-based knowledge Repository for IntegrateD Assessments" model 150 version 2.1 (FRIDA v2.1) and begin the process of evaluating an aggregated CHANS-driven modelling approach for representing the combined processes of climate and humans. FRIDA is a highly aggregated, global-scale, feedback driven model of the co-evolution of the world-Earth system. In Section 2, we further describe the modelling method applied, situating it among other methods and models that follow from those methods. Section 3 provides a top-level model description of FRIDA v2.1, describing the key processes represented for unifying the grand system-wide feedback between 155 the climate and human system. In the remaining sections, we evaluate the baseline results of FRIDA against the present state-of-the-art models, and importantly, demonstrate the level of precision that can be achieved by employing our alternative modelling approach.

## 2 What is FRIDA?

145

FRIDA is a global model that focuses on closing the system-wide feedback loops (processes) that cut across the climate and 160 human systems (see Schoenberg et al., (2025) for full model source code). In modelling the co-evolution of human processes contributing to emissions and the climate processes that transform emissions into climate change, FRIDA seeks to fully integrate the purposes of both traditional process-based IAMs and ESMs, allowing policy makers to simulate coherent policy scenarios. To achieve such an endeavour, while remaining computationally efficient, FRIDA prioritizes feedback complexity over specificity. By feedback complexity, we refer to the complexity arising from the dynamic interactions of interconnected feedback processes within the human-climate system that endogenously produces system behaviour (Senge, 2006). This is contrasted with the complexity arising from the sheer number of components (whether they be spatial or sectoral disaggregations) and fine-grained details within the system (which we refer to as specificity). Unlike ESMs that operate at fine-grained spatial scales, FRIDA foregoes specificity by being highly aggregated and not spatially resolved. While still Deleted: other Deleted: integrated assessment models are starting to do, Deleted: This Deleted: Deleted: the alternative Deleted: Deleted: the Deleted: Deleted: this Deleted: Deleted: alternative

Deleted: an alternative approach to the status quo Deleted: alternative

Deleted: Deleted: the alternative modelling approach Deleted: adopted in model construction

Deleted: approaches Deleted: and models

Deleted: alternative

Formatted: Font: 10 pt Formatted: Font: 10 pt Formatted: Font: 10 pt Formatted: Font: 10 pt **Field Code Changed** 

process-based, FRIDA departs from more highly detailed IAMs by seeking to represent the minimum level of detail required to model human behaviour endogenously, with the goal of closing the essential human-climate feedback loops, foregoing regional and/or sectoral breakdowns. As an illustration, we identified the essential sources of climate feedback using Technical Summary to the WGII report of the IPCC AR6 (Pörtner et al., 2022). We then categorised the sources of climate feedback into three broad areas: those necessary for prioritised inclusion within FRIDAv2.1; those which could plausibly be represented in a model such as FRIDA, but which reflected a lower priority, whether due to the perceived magnitude of their effect or the timescale and detail required to facilitate their implementation; and finally those whose inclusion in a highly aggregated model such as FRIDA was not deemed feasible. These judgements were made by the authors and their networks of subject matter experts. More detail on these sources of climate feedback can be found in Wells et al. (2025).

This approach allows for the endogenous generation of model behaviour, rather than relying on external inputs about climate, people, or goals for integrated assessments of baseline and policy scenarios. Because of the relative simplicity of FRIDA (especially when compared to ESMs and highly detailed process-based IAMs or the optimisation procedures required for cost-benefit IAMs), fewer computations are needed to obtain results from the model. One run of FRIDA from 1980-2150, with our timestep of 1/8th of a year, computed using Runge-Kutta 4 integration, takes only a few seconds on a current consumer grade laptop; this means when using the same computational resources that ESMs or many IAMs require to produce results, we can instead run FRIDA many times over, allowing large uncertainty ensembles to be produced, enabling us to measure and report confidence bounds within our single model and estimate uncertainites. We believe that measuring and reporting this uncertainty in IAMs is critical because it: (1) avoids giving a false sense of precision; (2) supports better decision making by making clear risks which may occur in worst/best case scenarios; and (3) builds credibility by transparently communicating what is and is not known. Figure 1 visualizes the distinguishing characteristics of FRIDA along the three dimensions (feedback complexity, specificity, and measurement of uncertainty within a single model), setting it apart from traditional IAMs and ESMs. The limitations of a globally aggregate, top-down modelling approach, as used in FRIDA, preclude much of the fine-grained fidelity that is available to policy makers today using existing IAMs to formulate climate policy. Although, we believe those tools may be sacrificing consistency and therefore accuracy, as a result of their regional and sectoral fidelity. This further includes the loss of interpretability as well as the loss of uncertainty measurement that results from additional specificity.

205

Deleted: ,

Deleted:

Deleted: as a result of their regional and sectoral fidelity

**Deleted:** as well as understanding relative to FRIDA and it's highly aggregated CHANS driven approach...

Figure 1: Plot showing how FRIDA relates to ESMs and Traditional IAMs (both process based and cost benefit) on the 3 axes of, feedback complexity, specificity and measurement of uncertainty. Shaded areas are used for ESMs and IAMs to represent the breadth of their development. FRIDA is represented as a solid line, since it is a single model. FRIDA has far less specificity, but in exchange is able to represent more feedback complexity. Its computational simplicity due to a relative lack of specificity allows for a deeper exploration of uncertainty as well. The specific values shown here are qualitative indicators of "generic" models of that type.

FRIDA employs the System Dynamics method (Forrester, 1961; Sterman, 2000), using ordinary differential equations to model the endogenous behaviour of the world-Earth system. The equations serve as direct representations of real-world processes included in the model based on our understanding of the human-climate system's functioning and supported by the best available scientific literature. The inclusion of each feedback process is predicated on the insights it provides and its potential for enhancing system understanding. Consequently, FRIDA embodies the laws of nature, well-established empirical relationships, as well as leading theories from economics, environmental psychology, and other relevant fields. To build confidence in FRIDA's outputs, we perform an iterative process of behavioural and structural validation throughout model construction (Barlas, 1996; Wilson et al., 2021). For behavioural validation, we assess FRIDA's performance when

Deleted: relative

endogenously reproducing 158 observed time series, spanning the scope of the model. When available, best-estimate parameter values and uncertainty ranges from literature are used. When that is not possible, we use calibration to set parameter values within either literature determined, or subject matter expert decided ranges. This calibration process ensures that the model accurately represents measurable aspects of reality, while giving us the ability to track the uncertainty inherent in the model's calibration.

Concurrently with behavioural validation, the structure is constructed and validated by subject matter experts to ensure that the modelled processes align with known conceptualizations of the attendant real-world processes, producing the right behaviour for the right reasons. Other notable structural validation tests performed include dimensional consistency verification, where each parameter and equation in FRIDA is assigned a unit that is conceptually and mathematically consistent across the entire system; extreme conditions tests to ensure that the model responds appropriately under various scenarios, checking for the magnitude and directionality of change in relevant variables; and sensitivity analysis, to measure the range of plausible outcomes, reflecting parametric uncertainties in our calibration data and, to an extent, structural uncertainties in our model structure.

FRIDA is not an optimizing model, this means that with FRIDA we do not simulate perfectly rational economic behaviour. Instead, we simulate the expectation formation and adjustment process of people. The System Dynamics method offers a litany of techniques and tools to incorporate the processes of expectation formation and adjustment into models and we have made use of this rich literature (e.g. Barlas and Yasarcan, 2006; Cavana et al., 2021; Paich and Sterman, 1993; Sterman, 1987) in our representations of human behaviour throughout the FRIDA model.

# 3 Model description

FRIDA v2.1 models the human-climate system through seven top-level modules, as shown in Fig. 2. A module is a discrete unit of model structure, a sub-system of equations that can be run independently of the other modules if the necessary inputs are provided as exogenous data. We have constructed our model using modules to provide a clear organization of model scope and to enhance the transparency of the model's structure. These modules are chosen because they together capture the main sources of anthropogenic emissions that contribute to climate change and allow us to represent what are known to be the main impacts of climate change on humans. These modules interact with each other, forming a complex web of feedback loops even at this high-level of abstraction. Within the climate system, the model contains process-based representations of the carbon cycle, the Earth's energy budget, and the human-driven changes in the water cycle that impact both the carbon and the Earth's energy budget. On the human side of the world-Earth system, the model incorporates process-based representations of demographics, economics, agriculture, and energy for human use, along with human behaviour and technological change embedded within those subsystems. Each subsystem is internally represented and hard coupled with all the other subsystems, maintaining the minimum necessary detail to capture the maximum number of relationships to other

Deleted: ,
Deleted: but
Deleted: i
Deleted:

subsystems. The rest of this section describes the key processes represented in FRIDA v2.1. In addition to the high-level descriptions in this paper, forthcoming publications will delve into the details of each of these modules.

Figure 2: The seven high-level modules which make up the FRIDA model. The climate module and its impacts on the rest of the model are represented in dark green. The human modules and their relationships are represented in module specific colours.

While outside of the scope of this model description paper (where we do not discuss policy runs), policy analysis in FRIDA can be done by changing parameters or enabling new policy structures. Such changes impact the feedback mechanisms represented within FRIDA and represent 'what-if' policy experiments, Energy policy in FRIDA gives the end-user control over energy taxes and subsides, including a Carbon Tax. Land use policies give control over forestation, irrigation, and non-agricultural water use. Economic policy gives control over austerity of governments including debt to GDP ratio, central bank inflation and unemployment targets, sea level rise adaption spending measures, as well as taxes on profits, wages, and wealth. The final area of policy available in FRIDA are around demand side behaviours including food and energy demand,

Deleted: perspective

Deleted: no

Deleted: ;

Deleted: ;

Deleted: which

Deleted: discussed below

Deleted:

as well as diet shift. It is important to note, that except for explicit dietary, and energy demand overrides, all these policy measures simulate the reaction of the human actors to the implementation of these policies and do not override endogenous behaviour. Furthermore, taxes and subsidies affect government budgets; subsidies are not free money but will necessitate adjustments to government spending elsewhere to avoid excessive debt.

# 3.1 FRIDA's climate processes

The representation of the climate system in FRIDA is based on a modified version of FaIR v2.1(Leach et al., 2021; Smith et al., 2018), modified in three key ways. Firstly, we replaced FaIR's carbon cycle emulator with our process-based land and ocean carbon cycle representation described below. The processes related to the carbon cycle are depicted in Fig. 3 and encapsulated within the top-level Climate and Land Use and Agriculture modules. Secondly, we reduced the number of anthropogenically-driven climate forcers substantially, retaining only those which were modelled explicitly within FRIDA's other modules. And finally, because of the structural changes we recalibrated our version of FaIR in its entirety following the published calibration procedures (Smith et al., 2024). Each sub-module depicted here represents a subset of the processes which we refer to as the climate system. These are processes that take input from, and greatly affect, the human processes within the FRIDA model.

**Deleted:** It is important to note, that with the exception of explicit dietary overrides and afforestation level, all of these policy measures simulate the reaction of the human actors to the implementation of these policies and do not override endogenous behaviour. Furthermore, taxes and subsidies affect government budgets; subsidies are not free money but will necessitiate future increases of taxes to avoid excessive debt

Figure 3: The <a href="mailto:sub-modules">sub-modules</a> making up FRIDA's representation of climate, whereby all the processes within the Climate module are represented in green. The human processes of the model, and their relationships to these climate processes, are represented in black. The Land Carbon module in green is a sub-module of the Land Use and Agriculture module; all other green modules are sub-modules of the Climate module.

The role of the Climate module in FRIDA is to simulate the Earth (climate) system's response to imposed forcings on the global scale. Population, GDP per capita, energy production, concrete production, food production, and land-use changes are the inputs to the climate module from the human portions (other modules) of the model. These inputs are used to calculate the global emissions of greenhouse gases (GHGs) and sulfur dioxide, changes to aspects of the land carbon cycle which result in changes to the land carbon sink, as well as changes in the land albedo and freshwater use. Likewise, these changes to the land carbon cycle are responsible for the carbon portion of Agriculture, Forestry, and Other Land Use (AFOLU) emissions within the Emissions sub-module. The Radiative Forcing sub-module calculates the resultant GHG concentrations in the atmosphere which drive in part both the Ocean's uptake of carbon, and the radiative forcing which is fed to the Energy Balance Model to calculate the requisite temperature response. The ocean uptake of carbon is represented within the Ocean Carbon sub-module, which takes as input both the temperature response from the Energy Balance Model, and the atmospheric CO<sub>2</sub> concentration, and feeds both back into the Radiative Forcing sub-module via the sea-air CO<sub>2</sub> flux

which is an important determinant of the CO2 concentration in the atmosphere. Finally, the globally aggregated dynamics of the Earth's cryosphere and water storage are simulated, allowing for the calculation of sea level rise.

There are four factors calculated directly from outputs of the climate model which are used as inputs in the processes 330 which represent the role of humans in the combined world-Earth system. Those four factors are: climate indices with record breaking exposure per person per year (climate extremes), heating & cooling degree days, sea level rise, and surface temperature anomaly (STA).

To produce those four factors which impact the processes representing the human system, FRIDA's climate representation includes eight chemical species of emissions. Of those, CO2, CH4, N2O, SO2 and HFCs represented as HFC-134a-equivalent are the five primary species, with emissions calculated directly from human activity. The three others (CO, NOx, and VOC) are modelled endogenously using other species as inputs to functions regressed from historical relationships. These eight species of emissions are translated into eleven different sources of forcing, nine of which are anthropogenic. The two sources of natural forcing include forcing from variations in solar radiation, and volcanic forcing, both provided as exogenous timeseries. The nine anthropogenic sources of forcing are CO<sub>2</sub>, CH<sub>4</sub>, N<sub>2</sub>O, Stratospheric Water Vapor due to CH<sub>4</sub> oxidation, Minor GHGs (HFC-134a equivalents, and Montreal Protocol Gases), Ozone, Black Carbon on Snow, Land Use, and Aerosols. The Aerosol forcing is further subdivided into aerosol-cloud and aerosol-radiation interactions. The structures we use to represent these forcings are equivalent to those of FaIR v2.1 for CO2, CH4, N2O, Stratospheric Water Vapour, Ozone, and Black Carbon on Snow. Minor GHG forcing is treated in a simplified manner due to the large number of these species in FaIR v2.1; Aerosol forcing is identical except for the inclusion of only one precursor 345 species (SO<sub>2</sub>); Land Use forcing is treated with greater process detail than in FaIR v2.1 (which assumes a forcing linear in cumulative AFOLU CO2 emissions), utilising the Land Use module in FRIDA. The set of ten forcers are necessary to represent the full human influence on climate which drives change in the energy balance model. The energy balance model used in FRIDA is the same as FaIR v2.1's three-layer system of equations representing the heat exchange between the deep ocean, the thermocline ocean, and the land & ocean surface.

The carbon cycle model in FRIDA exhibits a more advanced development compared to that of FaIR, owing to its process-based representation of oceanic and terrestrial carbon sinks. While FaIR simulates the atmospheric decay of CO2 emissions through a four-box model with varying decay lifetimes, FRIDA incorporates detailed process-oriented mechanisms, as elaborated below. The processes we have chosen to represent in FRIDA involve modelling the carbon sources and sinks related to the natural land carbon cycle, as well as land use and land use changes, specifically between the following categories: forests (maturing and mature), grassland, cropland and degraded land, in addition to each land use type's associated soil carbon pool. The soil carbon pools are modelled at a high-level using processes aggregated from LPJmL (Schaphoff et al., 2018) and are divided into two general categories: the fast and slow soil carbon pools. Human and naturally driven changes to the land affect not only the net primary production (per land use type) but also the soil carbon pools for each land use type. It is the movement of carbon between these states that is responsible for FRIDA's endogenous representation of the terrestrial carbon balance. While discussed here in the climate portion of the model, this logic ultimately resides in the Land Use and Agriculture module (Section 3.2.3) as this module ultimately represents the biosphere in FRIDA.

In a manner analogous to our approach for the terrestrial component of the carbon cycle, we apply a process-based four-stock ocean carbon model, incorporating comprehensive carbonate chemistry, to the marine component. This model encompasses the warm surface ocean, cold surface ocean, intermediate depth ocean, and deep ocean, and is primarily based on the ocean carbon cycle models developed by Lenton (2000) and Zeebe (2012). This is necessary to appropriately capture the ocean's response to increasing atmospheric CO<sub>2</sub> concentrations, and hence their role as a carbon sink.

Total global sea-level rise (SLR) is modelled as the sum of five different components, using the FRISIAv1.0 model (Ramme et al., 2025). FRISIA directly explicitly simulates the individual components of SLR as they have different drivers and can have regionally different impacts (Slangen et al., 2017). The five sources of SLR in FRIDA are: thermosteric, mountain glacier, Greenland ice sheet, Antarctic ice sheet, and land water storage. The land water storage component results from groundwater use as there is more demand for freshwater, particularly as more and more cropland is irrigated, and this (original) groundwater eventually ends up in the ocean, but at the same time this process is partly counteracted by additional water being stored on land in hydropower dams. The other three ice-related components are driven by STA-driven melting. Finally, thermosteric sea level rise results from the ocean water itself expanding as it warms, as a consequence of physical properties of seawater.

#### 3.2 FRIDA's human processes

FRIDA's human system comprises six separate subsystems (modules), each focused on a specific subject area: Demographics, Economy, Energy, Land Use and Agriculture, Resources and Behavioural Change – see Fig. 2. The representation of the human system starts with the humans themselves, i.e. population (the Demographics module), then moves into the system that people have constructed to meet their needs and desires (the Economy module). From there FRIDA represents additional process-oriented details around the specific human economic activities which are most responsible for the relationships between humans and climate that underlie people's economic activity, including: land use change and agriculture (Land Use and Agriculture module), energy production (Energy module), and concrete production (Resources module). Finally, to better represent the role of individual people in creating the demand component of the economy, we modelled the psychological processes related to behavioural change, which is currently applied to diet shift and food demand. The processes and interconnections within and between the human system are directly impacted by and directly impact, the climate system.

### 3.2.1 Demographics

The role of FRIDA's Demographics module (Fig. 4) is to represent how the global population changes over time. Global population is an important driver of demand for goods and services that will ultimately generate emissions that may unfold in the future. Therefore, as seen in Figure 4, the Demographics module does not directly impact the climate; instead, only

Deleted: s

Deleted: The role of population in FRIDA is to understand

Deleted: how

Deleted:

through other human processes are emissions directly generated. Population is modelled as a continuous cohorting system (Eberlein and Thompson, 2013), using Stella Architect's (the tool used to implement FRIDA) conveyors to represent the dynamics of ageing along with age-group specific mortality (Conveyor Computation, 2025). The continuous cohorts are grouped into seven age categories: Infants, Aged 1-20, Aged 20-40, Aged 40-60, Aged 60-65, Aged 65-75, and Aged Over 75 for the purposes of simulating different mortality rates, and the impact of climate on those mortality rates; each age group ends at the value described minus 1/8th of a year which is the solution interval of the model. The age-specific cohorts are initialized using United Nations (2022) data of global population by age in 1980, and the mortality impacts were modelled using Bressler et al. (2021), with Chen et al. (2024) for the age partitioning; see Wells et al. (2025) for details on how those relationships were determined, calibrated, and validated. Births are based on a calculated fertility rate, the result of a calibrated regression using GDP per person, and female literacy achievement as inputs, the rationale being that with rising standards of living and female education, birth rates drop exponentially (Kirk, 1996; Lesthaeghe, 2010; Marquez-Ramos and Mourelle, 2019; Proto and Rustichini, 2013).

Figure 4: A representation of how FRIDA's demographics module interconnects with the rest of the FRIDA model. The rest of the human processes are in black, the climate system in green, and the demographics module in blue.

# 3.2.2 Economy

405

The purpose of FRIDA's Economy module (Fig. 5) is to simulate the global economic system as a stock-flow consistent monetary model of production, consumption, finance, and government activities. It uses a Schumpeterian framework (Schumpeter, 1950, 1983) to model a dynamic circular flow of income, involving financial, corporate, government, and household sectors. The financial sector sets investment levels based on profitability, lending standards and

bankruptcy dynamics, while the corporate sector invests to expand production, hiring workers as needed. Corporate income is distributed as wages, rents, profits, and taxes. The government funds its spending through taxes and borrowing, adjusting expenditure when public debt-to-GDP rises above a pre-determined threshold (subject to uncertainty). The household sector, split into workers and owners, allocates income to consumption and savings, which creates demand for further investments. The corporate and financial sectors invest a small share of resources in high-risk exploratory activities, boosting productivity but raising bankruptcy risks, leading to short-term job losses. However, these investments ultimately foster long-term employment and economic growth.

Figure 5: Representation of the high-level process aggregations in FRIDA's economy module. This diagram depicts how the economy module interacts with the rest of the model. The rest of the human processes are in black, the climate system in green, and the economy module along with its sub-modules are in orange.

The model's Schumpeterian growth dynamics captures both quantitative and qualitative aspects of economic development. Quantitative growth results from increased output driven by labour (shaped by demographic factors and productivity gains) and by the accumulation of capital through both public and private investment. In contrast, qualitative

development is fuelled by exploratory investments undertaken by existing firms and new market entrants. These innovative investments determine real growth potential, with inflation occurring when income growth surpasses this potential. While innovation is beneficial in the long term, it creates economic stress in the short term, as obsolete economic activities are outcompeted, and temporary unemployment is created.

FRIDA's economic module also accounts for climate change, which impacts economic growth through four main pathways. First, rising temperatures reduce labour productivity (Clarke et al., 2022; Dasgupta et al., 2021). Second, sea level rise (SLR) damages infrastructure and assets, increasing public costs despite defensive measures (Diaz, 2016; Intergovernmental Panel on Climate Change (IPCC), 2023; Vafeidis et al., 2008; Wong et al., 2022). Third, financial instability grows as investment failures rise, curbing economic dynamism and long-term development by reducing the propensity to invest in both established and innovative projects (Carattini et al., 2023; Feng et al., 2024). Finally, maintaining climate-affected public infrastructure strains government budgets, potentially leading to austerity measures and broader economic challenges (Avtar et al., 2023). These interconnected impacts underscore climate change's significant hindrance to economic growth. To read more about these climate impacts see Wells et al. (2025).

Sea level rise impacts and adaptation are inherited from the FRISIA model (Ramme et al., 2025), and tracking coastal assets and population, in a modelling framework that is based on the Coastal Impact and Adaptation Model (Diaz, 2016; Wong et al., 2022). FRISIA expands upon that work by aggregating information for the use in a global, feedback-based model (Ramme et al., 2025). Coastal communities can respond to sea level rise in three user-defined ways: no adaptation, retreat or flood protection. The resulting costs are connected to the corresponding component of FRIDA's Economy, module. Increasing damages from storm surges are translated into increasing investment failures. Increasing people's exposure to storm surges is translated into a reduction in worker productivity, as in the FUND model (Tol, 2007). The costs of forced or planned coastal retreat are translated into owner spending. The investment into flood protection and increased maintenance costs are added to public expenditure, and lastly increasing fatalities in storm surges reduces the global population, affecting the economy indirectly.

While the Economy module produces little direct impact on climate, its indirect impacts through the modules described in the following sections that characterize other human processes necessary to represent the meeting of specific (emissions generating) human needs and desires do produce the very large majority of anthropogenic emissions that drive outcomes in the climate system.

## 3.2.3 Land Use and Agriculture

FRIDA's Land Use and Agriculture module (Fig. 6) represents process-based specifics around humans' needs and desires for food (vegetal and animal) and other land-based products (i.e. agricultural products, forestry etc.), and tracks how those production processes impact the Earth system, especially the climate system, and the rest of the human system at large. This module serves to add detail to the high-level, non-sectoral, specific processes modelled in the <u>Economy module</u>, so that the known empirical relationships between the needs and desires of the human population for goods and services based

Deleted: economic

Deleted: While this module produces little direct impact on climate, its indirect impacts through the forthcoming modules' more detailed explanation of the processes necessary to represent the meeting of specific (emissions generating) human needs and desires does produce the very large majority of anthropogenic emissions which drive outcomes in the climate system.

Deleted: economy

ultimately on the land can be represented. The Land Use and Agriculture module closes feedback loops with the climate system, and the more highly aggregated economic system via the supply demand balance for crops, and animal products, as well as the amount of various agricultural inputs including land nutrients (fertilizer), freshwater used for irrigation, and land inputs to produce the demanded agricultural goods which all ultimately either generate emissions or impact portions of the land-based carbon cycle or water cycle.

Climate

Climate

Crop

Crop

Freshwater

Land Use

Land
Nutrients

Land Use and Agriculture

Figure 6: Representation of the high-level process aggregations in FRIDA's Land Use and Agriculture module. This diagram depicts how the Land Use and Agriculture module interacts with the rest of the model. The rest of the human processes are in black, the climate system in dark green, and the Land Use and Agriculture module along with its sub-modules are in light green.

To represent the impacts of these processes, the Land Use and Agriculture module keeps track of the area of four types of land: forest (mature, and maturing), grass, cropland and degraded land. The corresponding vegetation growth is modeled as a function of climate and, in the case of cropland, additionally of agricultural management, thereby modeling crop productivity endogenously. The productivity aspects of the cropland are modelled within the Land Use and Agriculture module, as this land productivity represents the food, feed, seed, lost crops, and energy biomass demanded by the population and supplied by farmers (of all kinds) (FAO, 2024a, b). For all the other land use types, net primary production is modelled within the Land Carbon sub-module as documented in the Climate module (Section. 3.1).

Changes in this module generally result from changes in the Food Demand <u>sub-</u>module. Here, the demand for food, feed, energy (biomass), seeds, and the associated crop waste is summed to produce the overall demand for crops. Apart from food and feed demand (determined by changes in the Behavioural Change module) and biofuel energy (computed in the Energy module), all other demands are driven by real GDP per person and population (Fukase and Martin, 2020; Tilman et al., 2011).

In the process of meeting overall crop and animal product demand, the Land Use and Agriculture module represents the resultant changes in the areas and the land use intensities of the various land types in the Land Use <u>sub-module</u>. Furthermore, changes in agriculture management, such as fertilizer and irrigation, cause changes in crop productivity, water use, and therefore emissions. Finally, climate change creates a variety of both positive and negative impacts on cropland productivity. Increased CO<sub>2</sub> fertilization drives an increase in yield, and increases in STA almost always result in negative impacts. These relationships were derived empirically based on Franke et al. (2020) – see Wells et al. (2025) for more details.

On croplands, plants (crops) are grown to meet each of the demand sources mentioned above. Some parts of the plants are consumed by humans or animals, some parts are used for other purposes (e.g. burned in the field) and the rest is left on the field. Inside of the Land Carbon sub-module, discussed in Section 3.1 as a part of the Climate module, this plant litter builds up the cropland soil carbon while the soil carbon also decomposes, resulting in CO<sub>2</sub> emissions. Also represented within the Land Carbon sub-module, humans disturb the balance of natural land systems through forest land activities (timber production) and grassland activities (intensive animal grazing). Because of these activities, less carbon enters the soil carbon reservoirs. The soil carbon of these land-use systems is often much smaller than natural systems. If land-system change occurs, with an area of forest or grassland becoming cropland, the soil carbon reservoir decreases over time, including further emissions as from deforestation, leading to net land-use emissions until a new balance is reached. Since land use change is recurring, because new cropland is needed and old cropland fallows, this process is restarted continuously, leading to land use change being one of the largest global sources of carbon emissions (Friedlingstein et al., 2024).

Deleted: is

FRIDA's land use and agricultural module contains a highly aggregated Animal Products <u>sub-module</u> which includes supplier responses to changes in animal products demand, as well as feed availability, and land available for grazing. The purpose of this module is to represent the role of animal product production on the climate through feed consumption, grazing, and aquaculture, also closing key feedback loops with fertilizer production, and grassland allocation.

The Freshwater <u>sub-module</u> is used for two primary purposes. The first is to track human changes to the water cycle, which contributes to sea level rise via the modelled ground water anomaly. Secondly, the role of irrigation in crop yield changes is represented. A key climate feedback is also contained within this <u>sub-module</u>, which drives down irrigation efficiency as STA rises (Allen and FAO, 1998; Guo et al., 2017; Monteith, 1965; Shi et al., 2020) – a process described in Wells et al. (2025).

Continuing with the impact of the Land Use and Agriculture module on climate, of particular concern is the role that land takes with respect to emissions of anthropogenic carbon and the uptake of anthropogenically emitted carbon; the efficiency of land and ocean in taking up carbon directly affects the amount of CO<sub>2</sub> that is left in the atmosphere and thus climate change. Both land use parts of the carbon budget are kept track of in FRIDA within the Land Carbon sub-module.

# 3.2.4 Energy

FRIDA's Energy module (Fig. 7) represents, in a process-based manner, humans' needs and desires for energy and those processes' effects on climate. This module serves to add detail to the high-level processes modelled in the economy to close the feedback loops between the energy needed to satisfy the needs and desires of the human population for goods and services and the climate system. This module further provides feedback to the more highly aggregated economic system via the supply demand balance for energy, the investments needed to produce the energy, the average marginal net cost of energy, as well as the taxes and subsidies involved in regulating the production of energy. The structure of the energy supply sub-module in FRIDA is derived from the Model of Investment and Technological Development (MIND) model (Edenhofer et al., 2005; Held et al., 2009). This includes learning by doing processes as well as resource scarcity effects. The structure differentiates between primary and secondary energy for fuel-based energy generation. While the MIND model represents two types of energy generating process, fossil (with resource extraction, primary and secondary energy) and renewable energy, the energy supply sub-module in FRIDA has been extended to cover three types of fossil fuels (coal, oil, gas), four types of renewables (biofuel, hydropower, solar, and wind), and nuclear energy. Biofuel is a special case bridging the fossil energy and renewable energy sectors. This is implemented by treating fossil oil and biofuel as substitutes in the secondary energy production from oil. While fossil oil is extracted from resources, biofuels are processed from agricultural products and are therefore linked to crop demand, crop production and all the other inputs to the agricultural process.

Figure 7: Representation of the high-level process aggregations in FRIDA's Energy module. This diagram depicts how the Energy module interacts with the rest of the model. The rest of the human processes are in black, the climate system in green, and the Energy module along with its first level-sub-modules are in yellow. The energy supply <u>sub-</u>module is further divided in sub-modules which represent each energy type individually (coal, oil, gas, solar, wind, hydropower, nuclear, and biofuel).

To represent the impacts of these human economic processes, the Energy module represents the consequences of the world's energy demand in terms of emissions, average marginal cost of producing energy using the existing technologies, and supply/demand imbalance. The Energy module determines the investments into the energy sector needed to satisfy energy demand based on the change in the supply/demand imbalance over time. This is implemented using a PID (Proportional–Integral–Derivative) controller (Lunze, 2020), ensuring that investments are adjusted not just in response to the current imbalance but also in response to accumulating small imbalances and in response to a rapid change in the imbalance. The investments into the different energy sources are allocated based on their relative marginal costs, though a winner-takes-all system is not used; instead FRIDA's investment allocation function simulates simultaneous investment into multiple energy sources with the priority set by least marginal cost. The costs of producing additional energy depend on the

Deleted: e

existing and past use of the respective energy sources through learning by doing, resource scarcity, and stepping on toes effects<sup>1</sup> (see MIND (Edenhofer et al., 2005; Held et al., 2009) for more details on these effects). To match past investment choices present in the calibration data, implied cost adjustments (taxes/subsidies or simply the effect of overruling preferences) for the historical period are applied. It is the allocation of these energy investments by source that ultimately determine market share, as energy sources with more investment are able to supply more energy to the market.

The Energy module represents the demand for energy on a per capita basis irrespective of its source. Energy demand is driven by GDP per person, with increases in GDP causing increased energy demand, but with a decreasing marginal effect. Both energy production and energy demand are affected by the Climate module. Energy demand increases with an increase of cooling degree days (the population weighted average number of days a year with a temperature above 22 °C) and decreases with a decrease of heating degree days (the population weighted average number of days a year with temperature below 18 °C). Energy supply is affected both by a general increase in the deterioration of installed capacity with climate change due to extreme events, as well as <u>by</u> changes in the production efficiency of thermoelectric and hydroelectric power plants because of changed water flow, with thermoelectric plants additionally affected by increased river temperatures. To understand more about how these processes are modelled, see Wells et al. (2025).

Of particular concern for climate is the contribution of the Energy module to emissions of climate-relevant chemical species. The combustion of fossil fuels and biofuels emits greenhouse gasses and SO<sub>2</sub> into the atmosphere, unless these are combined with storage solutions for CO<sub>2</sub>. These aspects of the Energy system are represented in the Emissions sub-module of the Climate module (Section 3.1).

## 3.2.5 Resources (concrete)

FRIDA's Resources (concrete) module's (Fig. 8) purpose is to represent the process-based specifics around human's needs and desires for concrete and the effect of the production of concrete on climate via emissions. This module serves to add detail to the high-level non-sector specific processes modelled in the economy so that the known empirical relationships between the needs and desires of the human population for concrete for buildings and infrastructure can be represented in a way which both closes the main feedback loops with both the climate, and the more highly aggregated economic system via the production and maintenance of concrete buildings and infrastructure.

<sup>&</sup>lt;sup>1</sup> The stepping on toes effect (Jones and Williams, 2000) represents a limiting function on the marginal productivity of investment in cases of simultaneous investments, in our case into energy capital. The effect includes the deleterious effect of duplicate research and development as well as cost increases from material bottlenecks that would arise under such situations.

Figure 8: Representation of the high-level process aggregations in FRIDA's Resources module. This diagram depicts how the Resources module interacts with the rest of the model. The rest of the human processes are in black, the climate system in green, and the Resources module in brown.

The production of concrete, and specifically cement, which is the primary binding agent of concrete, is responsible for about 4% of annual anthropogenic CO<sub>2</sub> emissions (Friedlingstein et al., 2024). The Resources (concrete) module simulates the global production and in-use stock of concrete and associated CO<sub>2</sub> emissions. Two types of uses are considered: concrete in housing and service buildings and concrete in infrastructure. These structures are divided up into two age categories each ('New' and 'Old') and as structures age, they transition from 'New' to 'Old' until they reach their estimated service lifetime by which they are decommissioned.

The demand for concrete based housing and infrastructure is modelled using population size and GDP per capita. As the population grows and/or economic output per person increases, the demand for housing and infrastructure increases, which drives further construction and hence concrete production and associated emissions (87-164 kg CO<sub>2</sub>/ton of concrete depending on use-type) (Swedish Concrete, 2022). Additionally, in-use concrete structures deteriorate over time and require maintenance/repairs that also increase the global concrete production. The rate of deterioration increases with climate change, requiring more concrete to be produced to maintain the existing stocks of concrete infrastructure, due to both to damages to coastal assets from extreme weather events (computed as a sub-fraction of asset storm damages in the Sea Level Rise Impacts and Adaptations sub-module within the Economy module) and in general from enhanced corrosion (scaling with STA) (Bastidas-Arteaga and Stewart, 2015; Stewart et al., 2011; Wang et al., 2012).

Deleted: Concrete

# 3.2.6 Behavioural Change

The purpose of FRIDA's Behavioural Change module (Fig. 9) is to model the process-based specifics which underlie individual human decisions around dietary behaviour. Dietary decisions, including caloric intake and diet composition (animal vs. vegetal products), are environmentally significant behaviours that contribute to intensified food production and the resulting GHG emissions (Friedlingstein et al., 2024). Moreover, meat consumption has strong social-cultural significance for many people, raising barriers to collective action for reducing consumption of animal products (Manfredo et al., 2017; Stoll-Kleemann and Schmidt, 2017). This makes dietary behaviour a fertile ground to explore and model the social-cultural dynamics of behavioural change. The modelling framework underlying the Behavioural Change module is informed by and integrates knowledge from behavioural theories, empirical scientific literature, and participatory modelling (e.g., Bamberg and Möser, 2007; Godfray et al., 2018; Hammerseng, 2024; Milford et al., 2019; Rajah et al., 2024; Rajah and Kopainsky, 2024, 2025; Shove, 2010; van Valkengoed et al., 2025).

Deleted: drive

Figure 9: Representation the high-level process aggregations in FRIDA's Behaviour Change module. This diagram depicts how the Behaviour Change module interacts with the rest of the model. The rest of the human processes are in black, the climate system in green, and the Behaviour Change module along with its sub-modules are in purple.

The framework represents the social-economic-cultural-environmental processes involved for endogenously modelling the daily average per capita food demand and consumption (Total Food Demand <a href="sub-module">sub-module</a>) as well as the daily average per capita animal products demand and consumption (Animal Products Demand <a href="sub-module">sub-module</a>). Vegetal products are included as the difference between the two. Both these <a href="sub-module">sub-module</a> provide inputs to the Land Use and Agriculture module for affecting production dynamics and, subsequently, the climate system. Importantly, the climate feedback is closed via the Climate Risk Perception <a href="sub-module">sub-module</a>, which models the perceived climate change risk formed by three inputs from the climate system: (1) the surface temperature anomaly, as a proxy for climate information regarding global warming that is reported; (2) exposure to record-breaking extreme weather events; and (3) exposure to SLR-induced flooding. In turn, the perceived risk impacts subsequent dietary decisions.

The desired per capita demand (behavioural intention) is modelled as a function of three main motivational factors. First, the perceived accessibility of the consumption behaviour in terms of socioeconomic determinants such as income and price (Godfray et al., 2018; Milford et al., 2019). Second, the dynamic descriptive norm that motivates conformity to changing social trends of the prevailing standard behaviour (Cialdini, 2007; Sparkman and Walton, 2017; van Valkengoed et al., 2025). Third, the personal norms, or individually-held standards, that people expect of their own conduct based on personal values, beliefs and attitudes (Kaiser et al., 2005; Niemiec et al., 2020; Schwartz, 1977). These norms are further shaped by information from the social-ecological environment, including the perceived social-cultural value attached to the behaviour, the perceived risk of overconsumption, and the perceived climate change risk (Bamberg and Möser, 2007; Berndsen and van der Pligt, 2005; Manfredo et al., 2017; Wong-Parodi and Berlin Rubin, 2022). However, past behaviour inhibits sustained behavioural change (Linder et al., 2022; van Valkengoed et al., 2025). The extent to which the desired per capita demand is realized depends not only on the relative importance of each behavioural motivation to individuals on average, but also on the time taken for the formation of new habitual behaviours. A comprehensive model description of this modelling framework and its quantification is documented in Rajah et al. (2025).

Using this framework, the Behavioural Change module represents the multiple endogenous processes that influence how people change their dietary behaviour dynamically while accounting for changing social-ecological conditions.

#### 4 Model evaluation and calibration

The effectiveness of the FRIDA model as a tool for evaluating the proposed alternative integrated assessment modelling approach relies on its ability to accurately replicate patterns of observed behaviour across both the climate and human systems, while also creating plausible trajectories for these patterns up to 2150. To generate results using the FRIDA model, we generate an ensemble of runs, each with a unique deterministic parameter set, to compute confidence bounds around projections. For this paper, we run 100,000 members in our ensemble. The likelihood of a trajectory generated by a given

parameter set of the model is defined as the likelihood that the residuals between the observed measurements (calibration data) and the model results based on the parameter set are independently and identically distributed with the same statistical properties as in the maximum likelihood case. This is the same basic assumption made in ordinary least squares. However, in the case of ordinary least squares an analytical solution exists for both the maximum likelihood values as well as the resulting uncertainty ranges. In the case of a model like FRIDA no such analytical solution is possible to produce, so the maximum likelihood parameter values must be determined numerically via calibration.

Calibrating the FRIDA model was done using gathered measurements where possible over the period of 1980 to 2023. The FRIDA model underwent a "validation/verification/calibration" process, as outlined by Walker and Wakeland (2011), to improve its calibration. Calibration was performed using Powell's (2009) BOBYQA algorithm, a gradient descent method implemented in Stella Architect 3.8 (Stella Architect, 2025), which utilizes the Dlib C++ Library (dlib C++ Library, 2017).

The calibration process involved partial calibration, whereby each of the modules described above in Section 3 was calibrated largely independently. The Climate module was not directly calibrated with the rest of the model; instead, where possible, published parameter estimates from the literature were used, as FaIR's calibration spans a longer timeframe (from 1750 forward). Any changes to the published parameter estimates from FaIR, including the ocean carbon cycle structure, were made by applying the FaIR calibration process (Smith et al., 2024) to the altered FRIDA Climate module, over the same, 1750 forward time period as FaIR's original calibration. Similarly, unaltered portions of the Energy module inherited parameters from the MIND model, its foundational framework.

Partial calibration was employed to manage computational complexity and to iteratively refine the model in isolated segments. This approach minimized the chances of the optimizer stalling in flat payoff regions and facilitated better interpretation of parameter estimates and data fits. The calibration process aimed to minimize the square error between observed and simulated data, with weights set so that the payoff value approximated the number of data points. This weighting ensured the objective function behaved like the negative log-likelihood, increasing compatibility with the BOBYQA algorithm.

The FRIDA model has been calibrated to reproduce 158 different time series spread across the entirety of the model's scope (see Schoenberg et al., (2025) for a full listing). To measure the uncertainty in the calibration, and therefore the precision of the simulation results, we performed a 100,000 member global sensitivity analysis (Saltelli et al., 2007) across all parameters in FRIDA which did not have exact definitions from the literature. There were 801 individual parameters directly varied in the sensitivity analysis. In addition to those 801, there were 59 parameters which needed to be sampled together based on the FaIR calibration process applied to the climate portion of the FRIDA model. 100 different combinations for those parameters were determined from the Climate module's calibration and were varied across the global sensitivity analysis. The same approach was used for any multiparametric climate impact function, except only 11 combinations were used since only two parameters were being co-varied.

The ranges for each parameter sampled in the sensitivity analysis were determined through a process which started with literature-based research. During the calibration process described above, for any parameter which did not have a Deleted: e.g..

Deleted: exists

Deleted: c

Deleted: in

published range, wide ranges were set based on the intuition of the subject matter expert building that section of the model. This posed a challenge when it came to the sensitivity analysis. Most parameter combinations generated from the exhaustively wide parameter ranges set via intuition produced simulation results which were entirely implausible (division by zero, numerical overflows etc). Therefore, to be computationally efficient, i.e. not to have to run billions (or more!) combinations of parameters to find model runs which were behaviourally plausible, the uncertainty sampling range for each parameter was reduced. Each side of each parameter's range was found independently, by doing single parameter uncertainty analysis, and halting range discovery either when the likelihood of the run produced fell below 1/1000th of the maximum likelihood that was found via calibration or the end point of the original wide range used during calibration was encountered. The ranges were then made symmetric using the minimum distance between an endpoint and the default value found using calibration. This was done to ensure that samples were equally spread around the default parameter set and that the ranges of highest likelihood were sampled. It is important to note that because the reduced parameter ranges were established assuming independence among the parameters, it could be possible to find a wider range for a parameter if it was co-varied together with other parameter(s). With that being said, the low tolerance (1/1000th of the maximum) for the individual parameter likelihoods means that at least the most relevant area of the parameter space has been sampled, and likely beyond that too.

Due to the large number of parameters, it was not possible to apply a grid sampling scheme to cover the input space. Even just three sampling points per parameter dimension would have resulted in 3801 combinations. Instead Sobol Sequence sampling (Sobol' and Levitan, 1999) was employed to best spread the 100,000 sampling points across all 801 parameter dimensions simultaneously (Burhenne et al., 2011). Due to the generally wide range for each parameter sampled, and the comparatively low density of sampling in the hypercube of parameter space, the many runs obtained had a relatively low statistical likelihood. If the results were weighted by these likelihoods only a relatively small number of runs would contribute to the determination of the uncertainty ranges in the results. Consequently, ensemble runs are not weighted by their likelihood but can still be interpreted probabilistically. These bounds indicate regions of the output space that were encountered most frequently and, thus, represents a probabilistic interpretation conditioned on our sampling scheme. It is important to note that the confidence bounds reported here are larger than what it would have been if likelihood weighting were to be done. Therefore, the simulation results pictured in this report show a larger uncertainty range than would be expected at an increased sample size of 1,000,000 (or more) weighted by likelihood even accounting for the potential of wider parameter ranges determined with covaried parameters.

Each of the 100,000 uncertainty ensemble members represent a scenario not unlike an SSP baseline run, because no additional climate mitigation policies are introduced. Each ensemble member is a possible combination of parameter values within the ranges described above. Taken with the structure of the FRIDA model they imply a given amount of radiative forcing. The parameters varied across these runs cover everything from how climate impacts the human system, to how the human system impacts itself, to how the climate system impacts itself, and how the human system impacts the climate

**Deleted:** with the understanding

Deleted: be expected

Deleted: as

system. These parameters cover technological development, the relationships which govern economic processes, social processes, as well as parameters which condition climate's response to emissions.

A key difference between a FRIDA ensemble member run and an SSP is that ensemble members do not include exogenous time varying changes in parameters. The model's parameters are fixed over all points in time, for a particular run. However, as described in Section 2 and 3 above, many values that would commonly be exogenous parameters in other models are the result of endogenous processes within FRIDA. This includes some of the narratives covered by the SSPs, 735 such as the cost of renewable technologies or dietary choices. Not covered in this endogenous ensemble are future changes in policy. This includes climate policy but also means that changes in processes not explicitly modelled (e.g. global trade, a process represented via a time varying assumption in the SSPs) are not included in the ensemble. This ensemble does not include any future actions of government that change, such as energy taxes and subsidies, because the scope of the FRIDA does not include the processes which generate these taxes and subsidies, only the impacts of those taxes and subsidies. Nor does this ensemble allow for assumed future discoveries to change available technologies or costs beyond what is endogenously modelled (FRIDA's endogenously modelled processes include technological discovery and cost changes, but not major breakthroughs e.g. nuclear fusion). A key assumption in FRIDA is that all processes modelled continue to function in accordance with the structure created, and that all processes not modelled continue operating as they always have regardless of any input parametric or policy (outside of the scope of this paper) to the model. We call the projections generated from the FRIDA model and this set of assumptions our Endogenous Model Behaviour (EMB) scenario and it will serve as our baseline for all future analyses.

Figure 10 depicts the behaviour of key variables from each of the seven modules in FRIDA except for the Resources (concrete) module. Plotted on top of this EMB scenario ensemble, is data from six other standard IAMs (AIM/GCE, GCAM4, IMAGE, MESSAGE-GLOBIOM, REMIND-MAGPIE, WITCH-GLOBIOM) which were available for all the five SSP-Baseline scenarios. These data were collected from the IIASA scenario explorer (Riahi et al., 2017) and are displayed so that comparison can be drawn between FRIDA's output and these other more traditional IAMs.

Deleted: which

Deleted: for instance

Deleted: was

Deleted: This

Deleted: was

Deleted: is

Figure 10: 100,000 member ensemble of FRIDA varying all parameters across their most likely ranges. The EMB scenario (60) ensemble is without future climate policy; Outputs are (a) Surface Temperature Anomaly in degrees Celsius (b) Real GDP Per Capita in Thousands of SUSD2005 per year, (c) Sea Level Rise in meters, (d) Population in billions of people. (e) Animal Products Demand in exa-calories per year. (f) Forest land in billions of hectares, (g) Renewable energy output in exa-joules per year. (h) Secondary Fossil energy output in exa-joules per year. (h) Secondary Fossil energy output in exa-joules per year. The FRIDA EMB scenario ensemble is in the shaded grey, the dark line is the median, darker area is the 67% confidence bounds, Calibration (Historical Data) for each plot is in red. Each SSP is in its own colour and each of the 6 models is represented with its own line style and symbol. Comparison data is not plotted for Sea Level Rise as that data is not a part of the IIASA scenario explorer database. Animal Products Demand is measured in dry matter weight in the IIASA scenario explorer, and we could not reliably convert the unit to calories for comparison.

## 5 Discussion

As stated, among the key goals of this paper is to demonstrate the level of precision that can be achieved by employing our modelling approach, which sacrifices sectoral and spatial specificity to better represent the dynamic complexity inherent in the relationship between climate and humans. Figure 10 shows a range of simulation outputs covering a wide breadth of

uncertainties, thereby demonstrating that it is indeed possible to track and convey the level of precision resulting from the application of such a broad modelling approach.

This paper started by asking the question: is the increased specificity of models that focus on one part of the world-Earth system, without giving equal respect to both climate, humans, and their interconnections (i.e. ESMs and more loosely coupled IAMs) coming at the expense of structural errors and scenario inconsistencies due to the lack of an explicit, complete two-way coupling between climate and humans? Results from Figure 10 provide initial exploration of this inquiry.

First and foremost, if the structure of FRIDA, including its representation of climate feedback, is an at least somewhat accurate representation of reality, then, SSP1-Baseline (Van Vuuren et al., 2017), SSP2-Baseline (Fricko et al., 2017) and SSP5-Baseline (Kriegler et al., 2017) as represented by the six standard IAM implementations are not likely scenarios, and therefore should not be used to represent probable outcomes of inaction. For SSP5-Baseline, this is not a surprising outcome, as there has been critique of this scenario in the literature to date which is best summarized by Hausfather and Peters (2020). In their article, they also suggest using a moderate mitigation scenario from SSP2, the SSP2-4.5 scenario (among others) as a more plausible baseline scenario. While the EMB scenario ensemble contains within its 95% confidence bounds the warming characteristic of SSP1-Baseline, SSP2-Baseline, and for most of the 21st century, that of SSP5-Baseline, the high GDP per capita prescribed in any of those scenarios are incompatible with FRIDA's fully coupled structure, in the estimation of the authors this is likely due to the role of climate feedbacks included, both economic and non-economic, though SSP2-Baseline is the closest of the three. On the face of it, the six standard IAM model scenario results for each of these three scenarios present a development pathway to policy makers with consistently high economic growth in the face of a large warming, which our modelling in FRIDA deems implausible. This suggests that deliberately excluding two-way coupling between climate and humans within the six standard IAMs, as used to generate the SSPs, is important enough to significantly reduce growth in economic output as STA grows. Although, we must note that the lack of this structure is conceptually consistent with the protocol-necessitated exclusion of climate damages in the SSP framework, which has been suggested to limit their plausibility (O'Neill et al., 2014) especially for scenarios with high emissions (Van Vuuren et al., 2025). Further analysis is underway to confirm and explore this in more detail.

This leaves SSP3-Baseline (Fujimori et al., 2017) and SSP4-Baseline (Calvin et al., 2017) as the two scenarios most in line with the future projected by FRIDA's EMB scenario ensemble. The narratives behind these scenarios represent the most significant challenges for adaptation. By incorporating numerous feedback processes between climate and human systems, FRIDA appears to offer a more realistic, albeit sobering, assessment of the level of adaptation challenges that are likely to be encountered by humans. This of course is a natural outcome of having more, and stronger sources of climate feedback which require adaptation. For warming, FRIDA's EMB scenario ensemble encompasses both SSP3-Baseline and SSP4-Baseline within its 67% confidence bounds. For GDP per capita, both scenarios largely fall within the 95% confidence bounds. Unsurprisingly, because of the two-way coupling in FRIDA, both SSP3-Baseline and SSP4-Baseline align far more closely than any other scenarios, and this is why we believe they are likely more realistic baselines for policy making.

Deleted: loops

#### 6 Conclusion

835

840

This paper has introduced the FRIDA v2.1 JAM and demonstrated that its aggregated CHANS approach with its broad high-level, feedback focused modelling method can generate results with enough precision to be informative. This paper demonstrates a key benefit of an aggregated approach to CHANS modelling: it allows researchers to spend more computational and mental resources exploring the uncertainty which comes part and parcel with any modelling intended to support public policy making, while pointing a clear pathway towards the more likely scenarios that have large challenges to adaptation. Current state-of-the-art IAMs and ESMs have made a different set of choices to maximize the trade-offs between feedback complexity, specificity of model results (spatial and sectoral disaggregation), and the tracking and measuring of uncertainty. While the aggregated CHANS approach embodied by FRIDA is not capable of representing the same specific outputs of the existing ESM and IAM modelling community, it contains more climate impact feedback complexity, and is able to be used to calculate a more complete measurement of its uncertainty because the entire model takes only a few seconds on a single thread of a commonly available processor, so scaling it to tens of millions or even billions of runs is possible using the same specialized hardware that traditional ESMs and IAMs run on. FRIDA's role in the climate modelling community should therefore be to identify the most important climate/human interlinkages, and to clarify and communicate the uncertainty which underlies the modelling process which supports global climate policy making – uncertainty that is inherent in both natural and human processes, and critically relevant to their coupling!

Future work on FRIDA will be devoted to enhancing the representation of climate impacts, and carbon dioxide removal technologies, plus further disaggregation of energy supply and demand to make more policy intervention points available to model users. More importantly though, the focus of work will shift moving forward to model analysis, model-based reporting and stakeholder engagement. Further analyses done using this model will include studying the role of parameters in generating uncertainty, policy analyses, and structural analyses of feedback loop dominance, particularly as it relates to the climate impact structures we have implemented.

Code and data availability. FRIDA is released as a free and open-source model on GitHub <a href="https://github.com/metno/WorldTransFRIDA">https://github.com/metno/WorldTransFRIDA</a>.

The specific version used for this manuscript is available on Zenodo <a href="https://doi.org/10.5281/zenodo.15310859">https://doi.org/10.5281/zenodo.15310859</a> (Schoenberg et al., 2025).

The full infrastructure to run scenario ensembles with FRIDA is hosted on GitHub <a href="https://github.com/BenjaminBlanz/WorldTransFrida-Uncertainty">https://github.com/BenjaminBlanz/WorldTransFrida-Uncertainty</a>. Code for Figure 10 is available on Zenodo <a href="https://zenodo.org/records/15517312">https://zenodo.org/records/15396799</a> (Schoenberg, 2025).

Author contributions. Conceptualization: WS, CM. Data curation: CW, SM, JB, LR, JKR, MBG, BB, ANL. Formal analysis: WS, BB, JB, BC, MBG, ANL, JKR, LR, CW. Funding acquisition: CM, CS, WS. Investigation: WS, BB, JKR. Methodology: WS, BB, JB, BC, MBG, CL, ANL, JKR, LR, CS, CW. Software: BB, WS. Supervision: CM, CS. Visualization: WS, AM, JKR. Writing - original draft: WS, BB, BC, JKR, CW. Writing - review and editing: WS, BB, JB, BC, MBG, SM, AM, CL, ANL, JKR, LR, CS, CW, CM.

Competing interests. The contact author has declared that none of the authors has any competing interests.

Deleted: IAM, and

Deleted: approach

Deleted: our alternative (feedback first)

Deleted: feedback first

Deleted: true

Acknowledgements. This research was supported by the Horizon Europe research and innovation programs under grant agreement no. 101081661 (WorldTrans). The uncertainty analysis ensemble of the FRIDA model was run using resources of the German Climate Computing Centre (DKRZ).

Financial support. This research was supported by the European Union's Horizon Europe 2.5 – Climate, Energy and Mobility programme under grant agreement no. 101081661 (WorldTrans – Transparent Assessments for Real People). The funder was not involved in any part of the development of this research.

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
