# Peer review of "Introducing FRIDA v2.1: A feedback-based, fully coupled, global integrated assessment model of climate and humans"

_EGUsphere, 2025_

## Author Comment (AC1)

**Introducing FRIDA v2.1: A feedback-based, fully coupled, global integrated assessment model of climate and humans**

William Schoenberg, Benjamin Blanz, Jefferson K. Rajah, Beniamino Callegari, Christopher Wells, Jannes Breier, Martin B. Grimeland, Andreas Nicolaidis Lindqvist, Lennart Ramme, Chris Smith, Chao Li, Sarah Mashhadi, Adakudlu Muralidhar, and Cecilie Mauritzen

**Response to Referee #1**

This paper presents a system-dynamics model that couples the climate system with the drivers of anthropogenic emissions such as demographics, affluence, and energy-demand. This is an important modeling advance in my view, in that it adds structural diversity to the suite of models typically used to project emissions pathways (i.e. detailed process-based energy-system models with exogenously-prescribed trajectories of demographics, economic growth and, sometimes, technological change).

Key distinctions in this model that add value in my opinion are:

1. Endogenizing many processes that are either excluded or specified exogenously in most models used to asses future climate pathways. This includes both drivers of energy demand and emissions, as well as feedbacks from the impacts of climate change on those processes (e.g. impacts on heating and cooling demand, mortality, consumption growth).

2. A focus on quantifying parametric uncertainties. Because of the computational complexity of many detailed process-based integrated assessment models, results are rarely presented with comprehensive assessments of parametric uncertainties. The very different structure of this model allows a more comprehensive assessment of parametric uncertainties, which is key to placing model projections in an appropriate context.

Thank you for taking the time to review our manuscript and for your constructive feedback. Below we provide a point by point response along with the revisions we have made.

While adding significant value, it is important to note that these capabilities come with trade-offs. The manuscript notes the lack of regional differentiation which is an important limitation. I believe there are others though that should be more explicitly delt with in the paper, notably 1) the lack of any forward-looking belief formation or optimization, including for variables with a significant forward-looking component such as investments or fertility choices; and 2) the ability, at least in current form, to only model baseline behavior – there is no endogenous representation of collective action

or policy formation and no clear means of representing policy decisions given the endogeneity of key variables. Given these limitations, particularly the latter, I would appreciate if the manuscript could speak more clearly to intended either policy or research use-cases of the model.

We agree that these topics should be highlighted further within the paper. Regarding point 1), we have added the paragraph below the end of Section 2 , discussing the role of human behaviour and expectation formation in FRIDA.

"FRIDA is not an optimizing model; this means that with FRIDA we do not simulate perfectly rational economic behaviour. Instead, we simulate the expectation formation and adjustment process of people. The System Dynamics method offers a litany of techniques and tools to incorporate the processes of expectation formation and adjustment into models and we have made use of this rich literature (e.g. Barlas and Yasarcan, 2006; Cavana et al., 2021; Paich and Sterman, 1993; Sterman, 1987) in our representations of human behaviour throughout the FRIDA model."

Regarding 2), we chose to not highlight the implementation of variations in future policy in this paper initially, though it plays a key role within the FRIDA approach. We accept that clarification of this point will be of interest to readers, however. After Figure 2 we have now added the paragraph below which demonstrates the utility of FRIDA for policy analysis, though detailed discussion of this falls outside of the scope of this initial model description paper.

"While outside of the scope of this model description paper (where we do not discuss policy runs), policy analysis in FRIDA can be done by changing parameters or enabling new policy structures. Such changes impact the feedback mechanisms represented within FRIDA and represent 'what if' policy experiments. Energy policy in FRIDA gives the end-user control over energy taxes and subsides, including a Carbon Tax. Land use policies give control over forestation, irrigation, and non-agricultural water use. Economic policy gives control over austerity of governments including debt to GDP ratio, central bank inflation and unemployment targets, sea level rise adaption spending measures, as well as taxes on profits, wages, and wealth. The final area of policies available in FRIDA are around demand side behaviours including food and energy demand, as well as diet shift. It is important to note, that except for explicit dietary, and energy demand overrides, all these policy measures simulate the reaction of the human actors to the implementation of these policies and do not override endogenous behaviour. Furthermore, taxes and subsidies affect government budgets; subsidies are not free money but will necessitate adjustments to government spending elsewhere to avoid excessive debt."

We also edited the text in Section 2 to emphasize that policy scenarios can be run using this modelling approach and by extension in FRIDA.

"In modelling the co-evolution of human processes contributing to emissions and the climate processes that transform emissions into climate change, FRIDA seeks to fully integrate the purposes of both traditional process-based IAMs and ESMs, allowing policy makers to simulate coherent policy scenarios."

"This approach allows for the endogenous generation of model behaviour, rather than relying on external inputs about climate, people, or goals for integrated assessments of baseline and policy scenarios."

There are a number of additional comments I have on the current manuscript:

- Firstly, I note that evaluating the model as presented is challenging because only high-level information on model components is presented. Details of functional forms, parameters, and calibration for individual model components are referenced to other papers that are, in most case, still unpublished. Given these components may well change as part of that publication process, and because, until publication, these important details are unavailable for review, it seems it might be appropriate to wait on publication of the full model until those processes are finalized. I believe this is an editorial decision.

We respectfully disagree with the referee on this point. Publishing more detailed papers that outline the entirety of the equations for this entirely new model requires a paper like this. This paper discusses the philosophy and broad approach of FRIDA, as well as its validation, and calibration procedures – all of which those authors need a citation for. As a research team we decided the best way to break this chicken-and-egg problem was to produce a broad overview paper, demonstrating the approach under which FRIDA was developed so that authors publishing on specific portions of FRIDA can have an organizing framework they could write within while not having to repeat large sections of this paper in each of those papers. To produce an entire detailed documentation of FRIDA in a single paper would necessitate a paper of prohibitive length. Although, we have made direct reference to the full source code of the model in the first sentence of Section 2 where we introduce FRIDA.

"FRIDA is a global model that focuses on closing the system-wide feedback loops (processes) that cut across the climate and human systems (see Schoenberg et al., (2025) for model source code)."

- The current manuscript provides a high-level overview of model structure and the results of a full sensitivity analysis to parametric uncertainty. I believe the reader would benefit from additional details to better understand model behavior and dynamics and the calibration process. For instance, some elements that would be useful:
    - A supplementary table listing details on the 158-time series parameters used for calibration (e.g. reference, start and end date, variable, component)

We have made clear reference to the Zenodo repository which contains this information.

    - A better sense of key pathways, feedbacks or parameters driving model results. I recognize there are a complex set of relationships in the model, but it does not seem unreasonable to highlight essential feedbacks for the reader operating either within or between model components.

Ultimately, we feel this concern would be better discussed in the domain specific papers which will come, each interpreting the feedback origins for behaviour within the context of that specific sub-system. For an example of one such paper accepted for publication in GMD, e.g. see Rajah et al., 2025. We believe that we have adequately described the full set of connections between modules in Section 3. In that section, we have been sure to dedicate a portion of each sub-section and each figure to the mechanisms by which that module impacts the others, including Figure 2 which shows the full scope of the feedback present within the model. While we agree that we have not really touched on a feedback dominance analysis (the specific role of those feedbacks in the generation of behaviour in Section 5), we think that this would introduce another significant chunk of scope into this paper, muddying our purpose. Nevertheless, we have tried to, in part, ameliorate the referee's concern by highlighting the role of climate feedback in creating the results we present. The more explicit we get about the role of specific feedbacks, the more important it will be to present results and specific equations that defend those assertions. Consequently, it would make this paper less about the general approach of the FRIDA model; instead, it would delve into the specifics of the implementations of the feedback processes that we discuss – which we feel is better represented in the forthcoming model description papers of each module.

    - Similarly, a better sense of model sensitivity to parameters. This seems relatively easy to do given existing results – what parameters

(or interactions between parameters) are drive variance in key output variables?

While we agree with the referee that this would make for an interesting analysis, it does not fit within the purpose we intend with this paper – which is to present the approach by which FRIDA was developed and to point out what we see as the major consequences for the lack of two-way (CHANS) coupling in the SSP scenario creation process, and how that may impact the next round of the ScenarioMIP process. We have added an explicit mention of this analysis to our future work discussion.

"Further analyses done using this model will include studying the role of parameters in generating uncertainty, policy analyses, and structural analyses of feedback loop dominance, particularly as it relates to the climate impact structures we have implemented."

I find the interpretation of the set of 100,000 ensemble members quite confusing and believe this needs to be more clearly explicated. Page 25 implies that, given the lack of likelihood weighting in the sampling scheme, this set should not be interpreted probabilistically. But Figure 10 clearly suggests a probabilistic interpretation (i.e. describing 67% and 95% confidence bounds). If sampling is not likelihood weighted then can we really interpret these as confidence intervals?

- o One suggestion is that a comparison with probabilistic distributions of outcome variables (e.g. temperature, population, per-capita consumption) developed for Social Cost of Carbon purposes in Rennert et al. (2022) would be interesting. Unlike the SSPs, these are explicitly probabilistic ensembles, and so in that sense more comparable to the FRIDA output presented in Figure 10.

While it is true that we have not likelihood weighted our results in Figure 10, this does not mean that a probabilistic interpretation of our results is not possible. The confidence bounds shown in Figure 10 reflect the empirical distribution of outputs across the sampled input parameter space. These bounds indicate regions of the output space that were encountered most frequently and, thus, represents a probabilistic interpretation conditioned on our sampling scheme. As discussed in the paper, the current scheme overemphasizes the uncertainty of the model, which implies that the confidence bounds we provide encompass any likelihood weighted bounds. This makes us more comfortable with our presentation of confidence bounds. In order to clarify this point, we have expanded upon our discussion of the lack of likelihood weighting and on the interpretation of our confidence bounds by adding the following:

- "Consequently, ensemble runs are not weighted by their likelihood but can still be interpreted probabilistically. These bounds indicate regions of the output

space that were encountered most frequently and, thus, represents a probabilistic interpretation conditioned on our sampling scheme. It is important to note that the confidence bounds reported here would be narrower if likelihood weighting were to be undertaken."

While the suggestion to compare to probabilistic distributions generated for Social Cost of Carbon purposes is interesting and useful for an important purpose, it does not serve our purpose in this paper to present the framework for the FRIDA model by critiquing the scenario generating processes outlined and followed by the IPCC to date.

- I believe the abstract and introduction could do a better job of explicating the current typology of different models and distinguishing the contribution of the new FRIDA model. For instance:
  - The paper repeatedly references "IAMs" but that is a very broad class of models, and it is not always clear what types of IAMs (or whether its all of them) that are being referred to. I believe the distinction in Weyant (2017) between detailed-process IAMs (e.g. GCAM, IGSM) and cost-benefit IAMs (e.g. DICE, FUND, GIVE) is relevant here. My sense is that the paper is mostly referring to the former, in which case it is helpful to be specific

Our intent is to critique the approach of both of these types of IAMs for different reasons. Therefore, what is missing from our introduction and abstract is an explicit mention of the kinds of climate impact that close the feedback loops we are most interested in – i.e. non aggregated economic climate consequences. We have added the following statement to our abstract to address this concern.

- "The current group of models assessed by the Intergovernmental Panel on Climate Change (IPCC) to produce their assessment reports lack endogenous process-based representations of climate-driven changes to human activities, especially beyond the purely economic consequences of climate change. These climate-driven changes in human activities are critical to understanding the co-evolution of the climate and human systems."

In addition, to our introduction, we have added the nuance of Weyant's terminology for IAMs and to make our critique clearer we have added the following two statements:

- "That disconnect can be seen in both detailed process-based IAMs and cost-benefit IAMs (for terminology see Weyant (2017)) as an under representation of the impact of climate on human systems beyond direct

or highly aggregated impacts on economic output. In addition, both classes of models struggle to represent human behaviour considering the impacts of these damages."

- "The trade-off for this increased level of specificity in the ESMs and IAMs is a lack of explicit modelling of the grand system-wide feedback processes. Capturing these processes requires a CHANS perspective for a more complete representation of climate impacts beyond direct economic damages. Such a representation fully couples the climate and the human world together – creating the feedback which locks these two subsystems into a synchronous co-evolution."

  - The paper and introduction draw a distinction between ESMs capturing the earth system and IAMs capturing human elements. But many IAMs include a representation of the climate system (e.g. IGSM uses the MIT Earth System Model, GCAM is often connected to HECTOR). Typically that representation is an intermediate complexity model similar to the FAIR model used in FRIDA. All cost-benefit IAMs include a model of the climate system in order to calculate the benefits of emissions reduction. I feel like this discussion needs more nuance and specificity.

We have elaborated on the statements we included in the introduction that discuss models like these. We are trying very carefully to not critique specific models (either IAM or ESM) as we feel that our critiques apply broadly across all models in this space. As mentioned above, a target for critique is the RCP/SSP/ScenarioMIP framework, which by purposeful design requires the development of models which can be run in a way that does not represent the role of climate on humans. Therefore, we have adjusted our introduction with the following statements to capture what we perceive to be the referee's intent.

- "Global scale ESMs and IAMs are generally not directly coupled; instead, information from one is fed into the other, whether directly via exogenous inputs (as is typically done in ESMs to represent the future development of the human system), or indirectly via emulation (as is often done in IAMs to represent the response of the climate system to potential future human behaviour)."

- "For the class of IAMs that do contain process-based climate representations either via the inclusion of reduced-complexity climate emulators or by directly hard-coupling to ESMs, the RCPs and SSPs are used to standardize assumptions across scenarios and comparisons to other models, in effect perpetuating the inconsistencies from the uncoupled models to the coupled models."

We also direct Referee #1 to the comments made by Referee #3 which are similar in intent, where we discuss CHANS, and therefore raise and discuss specific models which do couple human and natural systems.

Bibliography

Rennert, K., Errickson, F., Prest, B. C., Rennels, L., Newell, R. G., Pizer, W., Kingdon, C., Wingenroth, J., Cooke, R., Parthum, B., Smith, D., Cromar, K., Diaz, D., Moore, F. C., Müller, U. K., Plevin, R. J., Raftery, A. E., Ševčíková, H., Sheets, H., … Anthoff, D. (2022). Comprehensive evidence implies a higher social cost of CO2. *Nature*, *610*(7933), 687–692. https://doi.org/10.1038/s41586-022-05224-9

Weyant, J. (2017). Some contributions of integrated assessment models of global climate change. *Review of Environmental Economics and Policy*, *11*(1), 115–137. https://doi.org/10.1093/reep/rew018

**Response to Referee #2**

**General Comments:**

The model presented in this paper is a feat of scope that demonstrates the feasibility of integrating endogenous feedback between climate and human processes. Comparing results of simulation ensembles to various SSPs highlights key differences and underscores the importance of including coupled climate-human feedback models in future IPCC assessments.

The FRIDA model is described through an interconnected set of domain-specific modules that represent different aspects of the system: climate, demographics, economy, land use and agriculture, energy, resources, and behavioral change. These modules are not strictly necessary from a computational perspective, but they are helpful for communicating the scope and feedback complexity of the model.

Thank you for the positive comments as well as the detailed list of corrections to be made. We have documented the changes we have made following your comments below.

**Specific Comments:**

In the introduction, lines 82-83, please elaborate on (define or explain) the named taxons.

We have done so using parenthetical notation after the introduction of each taxon. See the following for the revision.

- "First and foremost, the division between climate processes and human processes across models must be bridged. In the language of Donges et al. (2021), such an approach must represent the Biophysical taxon (i.e. the "natural laws" of physics, chemistry or ecology), Socio-cultural taxon (i.e. human behaviour and decision making), and the Socio-metabolic taxon (i.e. material interactions of the biophysical and human systems), with the necessary complexity and detail to capture the unique contributions of each set of interconnected dynamics to the evolution of the entire world-Earth system"

In the last paragraph of the introduction, the emphasis on approach (lines 97 and 99) implies a methodological orientation, but it seems that the major contribution is the novelty of the integrative model structure. This emphasis on approach could be reframed or clarified (invoking "approach" in a broader sense, not specifically

"methods") to temper expectations about what follows (such as that section 2 will be a "methods" section).

We've reframed the introduction to Section 2 here to say

- "In Section 2, we further describe the modelling method applied, situating it among other methods and models that follow from those methods."

My advice on reframing "approach" notwithstanding, I'd be interested in knowing more about the method by which the model structures were generated. For instance, in reference to line 115, it would be helpful to know more about how the "essential" loops were identified.

We have added the following text to the statement identified which clarifies the process by which the essential feedback loops were determined.

- "For example, we used the Technical Summary of the WGII report from the IPCC AR6 (Pörtner et al., 2022) to identify the key sources of climate feedback. These were then grouped into three categories: (1) those warranting high-priority inclusion in FRIDAv2.1; (2) those that could reasonably be represented in a model like FRIDA but were assigned lower priority, either because of their anticipated impact or the timescale and detail needed for implementation; and (3) those considered infeasible to include in a highly aggregated model such as FRIDA. These classifications were based on the judgement of the authors in consultation with their networks of subject matter experts. Further information on these climate feedback sources is provided in Wells et al. (2025)."

The paper does make some mention of how the structure was constructed and we have added a footnote to provide more explanation.

- "Concurrently with behavioural validation, the structure is constructed and validated by subject matter experts to ensure that the modelled processes align with known conceptualizations of the attendant real-world processes, producing the right behaviour for the right reasons."

In section 2, the term "relative simplicity" (line 117 and caption for Figure 1) could be confusing after emphasizing the richness of FRIDA's feedback complexity. It would be helpful to clarify what is meant here, perhaps by referring to the number of variables or using the phrase "relative computational simplicity."

Agreed. We reference the model's computational simplicity and how that is enabled by its relative lack of specificity.

In section 3, before getting into each of the specific modules, it would be helpful to explain what modules are and why they were used.

We have defined and described what modules are immediately after the first use of the word in the beginning of section 3

- "A module is a discrete unit of model structure, a sub-system of equations that can be run independently of the other module if the necessary inputs are provided as data.  We have constructed our model using modules to provide a clear organization of model scope and to enhance the transparency of the model's structure."

For the benefit of clarity, as the modules are presented, it would be helpful to be consistent about using the terms "module" to describe the modules at the highest level of aggregation and "sub-module" to describe the modules within the highest modules, even if they contain further sub-modules. Some of these instances are noted under Technical Corrections below.

We have clarified our language in all places, beyond those mentioned between a top-level module, and a sub-module.

In section 3.2.1, it would be worth pointing out in the context of Figure 4 that the Demographics module does not directly impact the Climate module, but rather that the feedback from population dynamics to the climate is mediated through other modules describing human processes.

Agreed, we have added the following two sentences to the beginning of 3.2.1 to clarify the role of population within FRIDA.

- "Global population is an important driver of demand for goods and services that will ultimately generate emissions that may unfold in the future.  Therefore, as seen in Figure 4, the Demographics module does not directly impact the climate; instead, only through other human processes are emissions directly generated."

In section 3.2.4, please clarify briefly what is meant by the "stepping on toes" effect (line 406).

We have added a footnote to explain the stepping on toes effect.

- "The stepping on toes effect (Jones and Williams, 2000) represents a limiting function on the marginal productivity of investment in cases of simultaneous investments, in our case into energy capital.  The effect includes the deleterious

effect of duplicate research and development as well as cost increases from material bottlenecks that would arise under such situations."

**Technical Corrections:**

Line 41: Suggest "processes that include but extend beyond the economic" instead of "processes both economics and not"

Corrected

Line 54: Suggest "represent known feedback" instead of "represent what is known to be the true feedback"

Corrected

Line 58: Suggest "has been proposed" rather than "is proposed" to be clear that the sequential approach is not being proposed in this manuscript

Corrected

Line 67: Suggest "has contributed to a division" rather than "has led to a division"

Corrected

Line 72: Should be "WG II and WG III" not "WG II or WG III"

Corrected

Line 105: Suggest "feedback loops" rather than "feedback processes" since "closing the loops" is more understandable than "closing the processes"

Corrected, put processes in parentheses to clearly draw an analogy for readers who aren't so used to thinking in terms of feedback to clarify that these are process based representations.

Line 119: Should be a semicolon after laptop, not a comma

Corrected

In Figure 1, the legend for ESM and IAM colors should match the colors used in the figure.

Corrected

In Figure 2, the arrow from the Demographics module to the Climate module appears to be in error. Such an arrow does not appear in the module-specific Figure 4.

Corrected

Line 181: In the caption for Figure 3, the first instance of "modules" should probably be "sub-modules."

Corrected

Line 190: Both instances of "module" should be "sub-module", as in "Emissions sub-module" and "Radiative Forcing sub-module"

Corrected

Line 203: Suggest adding "chemical" before "species", to read "eight chemical species"

Corrected

Line 254: should be "impact" not "impacts", as in "impact the climate system"

Corrected

Line 258-9: the citation (Conveyor Computation, 2025) is missing from the reference list

Corrected

Line 279: remove the extraneous word "becomes"

Corrected

In Figure 5, the orange arrow from the Economy module to the Climate module should be positioned at the edge of the orange shaded box to imply it is coming from the overall module and not from a specific sub-module like GDP.

Corrected

Line 309: capitalize "Economic" for consistency

Agreed, also changed to Economy for consistency.

Line 315-7: Suggest rephrasing of this sentence, mainly to avoid using the term "forthcoming" to refer to something elsewhere in the same paper: "While the Economy

module produces little direct impact on climate, its indirect impacts through the modules described in the following sections that characterize other human processes necessary to represent the meeting of specific (emissions generating) human needs and desires do produce the very large majority of anthropogenic emissions that drive outcomes in the climate system."

Accepted.

Line 322: Suggest specifying "Economy module" rather than "economy"

Corrected

Line 325: Suggest adding the words "closes loops" after "system" to read: "economic system closes loops via the supply demand balance"

We opted not to incorporate this suggestion as it would turn the full sentence into

- The Land Use and Agriculture module closes feedback loops with the climate system, and the more highly aggregated economic system closes loops via the supply demand balance for crops, and animal products, as well as the amount of various agricultural inputs including land nutrients (fertilizer), freshwater used for irrigation, and land inputs to produce the demanded agricultural goods which all ultimately either generate emissions or impact portions of the land-based carbon cycle or water cycle.

We feel that the double "closes loops" phrasing is more awkward, and the intent is to describe the ways in which the Land Use and Agriculture module closes feedback loops with the other key modules of the model, not how the economic system is connected.

Line 341: suggest "sub-module" not "module" in the first instance, as in "Land Carbon sub-module"

Corrected

Line 342: suggest "sub-module" not "module" in the second instance, as in "Food Demand sub-module"

Corrected

Line 347: suggest "sub-module" not "module", as in "Land Use sub-module"

Corrected

Line 353: should be "parts are" not "part is"

Corrected

Line 354: suggest "sub-module" not "module" in the first instance, as in "Land Carbon sub-module"

Corrected

Line 356: suggest "sub-module" not "module", as in "Land Carbon sub-module"

Corrected

Line 363: suggest "sub-module" not "module" in the second instance, as in "Animal Products sub-module"

Corrected

Line 367: suggest "sub-module" not "module"

Corrected

Line 369: suggest "sub-module" not "module"

Corrected

Line 374: suggest "sub-module" not "module", as in "Land Carbon sub-module"

Corrected

Line 416: add "by" before "changes" to read "as well as by changes"

Corrected

Line 419: suggest adding "chemical" before "species"

Corrected

Line 437: replace "Concrete module" with "Resources module" or "Resources (concrete) module"

Corrected

Line 456: remove extraneous word "drive" to read "barriers to collective action"

Corrected

Line 498: suggest removing "e.g.," before "ordinary least squares"

Corrected

Line 499-500: suggest adding "exists" after "solution" to read "an analytical solution exists" and removing it from the end of the sentence

Corrected

Line 524: remove extraneous word "in" to read "across the global sensitivity analysis"

Corrected

Line 567: suggest rephrasing with punctuation, to read "actions of government that change, such as energy taxes and subsidies, because"

Corrected

Line 578: should be "were" not "was"

Corrected

Line 579: should be "These data were" not "This data was"; and should be "are" not "is", as in "are displayed"

Corrected

Line 641: suggest removing the word "true" before "uncertainty"

Corrected

**Response to Referee #3**

This study presents a global coupled human and natural systems (CHANS) model built upon a system dynamics approach. In particular, the model incorporates various feedback that is not available in IAM models, and the results demonstrate the key role of this feedback. Also, I agree that SD-based models are important complementary tools to other models, including IAM and Earth system models. The manuscript could contribute to modeling CHANS dynamics and making future dynamic predictions.

Thank you for the positive comments as well as constructive criticism you have offered. We have documented the changes we have made following your comments below and the reasons why we have not made some of the changes suggested.

My major comment on this work is that it did not track the recent progress on the CHANS theory and modeling efforts well. The review process needs improvement, as many important work is missing in the current literature and discussion. There are classic papers about CHANS (Boyd 2017; Alberti 2011; Liu 2007), and more recent papers highlighting the need and challenges of two-way coupling for modeling CHANS (Motesharrei 2017; Li 2023). In terms of models, there are similar global SD models, such as Felix 2.0 (Ye 2024), ANEMI3(Breach & Simonovic 2021),iSDG (Pedercini 2019), and country/regional models, T21-China (Qu 2020), ANEMI_Yangtze (Jiang 2022), and the Yellow River (Sang 2025). What are the key differences between this work and other global SD models? Although regional models are different from global models in scale, the modules share many similarities. For IAM models, E3SM-GCAM is the latest coupled IAM and ESM models (Di Vittorio 2025). These existing modeling efforts often share similar challenges and difficulties as summarized in (Li 2023).

We thank the referee for showing us the extant literature on CHANS theory which we have included into the abstract, introduction and conclusion.

In the abstract we have made the following changes.

- "An alternative aggregated approach, which couples human and natural systems (CHANS) such as the one used to build the Feedback-based knowledge Repository for IntegrateD Assessments "FRIDA" v2.1 IAM documented here, integrates climate and human systems into a unified global model, prioritizing feedback dynamics while maintaining interpretability."
- "FRIDA demonstrates that an aggregate, feedback-driven modelling approach, capturing CHANS interconnections with rigorous measurements of uncertainty, is possible."

In the introduction we have made the following change.

- "Properly representing the co-evolution of the climate system with the humans who exist within it requires models that two-way couple climate processes with human processes that include but extend beyond economic dimensions (Calvin and Bond-Lamberty, 2018; Donges et al., 2017; Motesharrei et al., 2016). This class of models is referred to as CHANS models, which stands for coupled human and natural systems models (Alberti et al., 2011; Kramer et al., 2017; Liu et al., 2007). The current crop of models assessed by the Intergovernmental Panel on Climate Change (IPCC) to produce their assessment reports lack the CHANS perspective of endogenous process-based climate-driven changes to human activities (Beckage et al., 2022; Donges et al., 2021; Wilson et al., 2021)."

Throughout the paper, we have also reframed our "alternative modelling approach" as "an aggregated CHANS-driven modelling approach":

- E.g. "An aggregated CHANS-driven modelling approach that answers our question allows the SSPs to serve to categorize uncertainty in the presentation of future scenarios. Second, an aggregated CHANS-driven modelling approach requires a fully endogenous, process-based explanation for model behaviour. Without it, an aggregated CHANS-driven approach would not adequately address the main problem caused by current schism. Third, to increase the understanding derived from models built using an aggregated CHANS-driven approach, highly aggregated models are preferred (Robertson, 2021). Without these considerations, a model built following a more disaggregated CHANS approach risks building an opaque model which is far too complex to yield (actionable or trustworthy) insight."

And in the conclusion the following change was made.

- "This paper has introduced the FRIDA v2.1 IAM and demonstrated that its aggregated CHANS approach with its broad high-level, feedback focused modelling method can generate results with enough precision to be informative. This paper demonstrates a key benefit of an aggregated approach to CHANS modelling: ..."

Next, we thank the referee for reminding us of the extant System Dynamics based IAMs. We note however that we didn't ignore FeliX; we referenced FeliX (Eker et al., 2019) previously, but now we do so by name directly and we have added the suggested most recent reference for FeliX 2.0 (Ye et al., 2024). Next, we have also taken the suggestion from the referee to specifically mention ANEMI3, and E3SM-GCAM as examples of CHANS models, and therefore we have included En-ROADS for the purposes of discussing a more complete set. We have chosen not to include T-21 or its later iteration iSDG as well as ANEMI_Yangtze because those are not global models, and our scope in this paper is global models. Below is how we refer to existing CHANS models.

- "In turn, this begs the question: is the increased specificity of these ever more disaggregated models, both ESM and IAM alike, coming at the expense of structural errors and scenario inconsistencies being introduced from the lack of an explicit, complete two-way coupling between climate and humans? Global models which fall into the CHANS category including ANEMI3 (Breach and Simonovic, 2021), En-ROADS (Kapmeier et al., 2021), E3SM-GCAM (Di Vittorio et al., 2025), and FeliX (Eker et al., 2019; Ye et al., 2024) ultimately pose the same question although generally at higher levels of disaggregation, and therefore with challenges for interpretability."
- "Fully answering this question requires a model developed using an alternative, aggregated CHANS-driven approach that is complementary to the current ESMs and IAMs as well as the more disaggregated CHANS models."
- "Instead, as some of the more disaggregated CHANS models do, the thinking and narratives contained within the SSPs should be used to describe the potential for the future unfolding of human behaviour (see e.g., Eker et al., 2019; Ye et al., 2024)."

The model emphasizes its ability to provide endogenous feedback. For a model, it is also critical to have the capability of designing policy influence and external intervention scenarios to answer the "what if" question. A full endogenous model implies that it is not easy to implement scenario design. The potential user case of the model should be introduced.

Referee #1 made a similar comment, so we have included a paragraph in the beginning of Section 3 to describe the role of Policy in FRIDA and therefore how the model is used.

The paper emphasizes the inclusion of two-way feedback compared to one-way feedback in IAM. What processes and results are influenced by the newly added feedback loop?

We refer the referee to the comments and changes in response to Referee #1 who suggested a similar discussion.

Is there a way to quantify the effects?

Yes, there are ways to quantify the effects, but they have never been applied to a model of this size or complexity, and this is fundamental research we are working on for a future study. The sub-field of System Dynamics which studies how to do this is called structural dominance analysis, and specifically the method we are referring to is called Loops that Matter (Schoenberg et al., 2020). An objective quantification of the feedback loop dominance profile of FRIDA will require its own complete research paper to cover both the methodological advancements to Loops that Matter and the results

of the analysis of FRIDA. We make note of this in the final sentence of the paper where we refer to structural analyses of feedback loop dominance.

"Further analyses done using this model will include studying the role of parameters in generating uncertainty, policy analyses, and structural analyses of feedback loop dominance, particularly as it relates to the climate impact structures we have implemented."

The authors also discussed different results with IAM, but the exact causes of the differences should be elaborated.

Likely we will never truly know the specific parameters and structural differences which provably and ultimately cause the differences between FRIDA and the other IAMs. All the models involved are far too complex for that kind of a specific comparison. As we mention in Section 5, we attribute the differences to the role of climate feedback. We have amended section 5 to say the following to make our position clearer on the source of the differences.

- "While the EMB scenario ensemble contains within its 95% confidence bounds the warming characteristic of SSP1-Baseline, SSP2-Baseline, and for most of the 21st century, that of SSP5-Baseline, the high GDP per capita prescribed in any of those scenarios are incompatible with FRIDA's fully coupled structure, in the estimation of the authors this is likely due to the role of climate feedbacks included, both economic and non-economic, though SSP2-Baseline is the closest of the three."

Specific comments

L67-68. Check Li 2023

Citation added:

- "The disconnect in the modelling process has contributed to a division of responsibility for representing the co-evolution of climate and humans, making CHANS modelling a new frontier for global integrated assessment modelling (Li et al., 2023)."

L75-77 Check Motesharrei 2017

We thank the referee for this reference and have added it to the first statement in our introduction where we reference the importance of models which close feedback.

- Properly representing the co-evolution of the climate system with the humans who exist within it requires models that two-way couple climate processes with human processes that include but extend beyond the economic (Calvin and Bond-Lamberty, 2018; Donges et al., 2017; Motesharrei et al., 2016)

L93-100. There are many alternative models of IAM, including SD model and Agent-based models, ESM-IAM (Yang 2015). Each modeling method has its own pros and cons.

We agree that there are multiple approaches to modelling including, System Dynamics (the method we chose) Agent Based, Discrete Event Simulation among others, but we feel that on balance it would not justify the space if we were to present each to the reader of this paper with its pros and cons. The purpose of this paper, which we have expressed more clearly with the help of the comments from all the referees, precludes this kind of exploration of modelling methods not used.

L115-116. The lack of regional breakdown also limits the model's applicability to support real-world policy making, which is important for CHANS models.

We agree wholeheartedly with the referee about the difficulty in using FRIDA in a regional context but wish to point out that global models with a regional breakdown by and large lack the CHANS feedback, and those which do not, lack the potential for easy interpretability. Therefore, we argue that the approach used for FRIDA is valuable to the policy making community because it does contain that CHANS feedback, and because of its relatively high level of aggregation, it is simpler and easier to interpret. To communicate that, we added the following sentences to the end of the first paragraph in section 2.

- "The limitations of a globally aggregate, top-down modelling approach, as used in FRIDA, preclude much of the fine-grained fidelity that is available to policy makers today using existing IAMs to formulate climate policy. Although, we believe those tools may be sacrificing consistency and therefore accuracy as a result of their regional and sectoral fidelity. This further includes the loss of interpretability as well as the loss of uncertainty measurement that results from additional specificity."

L385-386: How to determine the share of each energy type

We added a clarifying statement to section 3.2.4 to describe how energy investments shape market share.

- "It is the allocation of these energy investments by source that ultimately determine market share, as energy sources with more investment are able to supply more energy to the market."

What is the time step of the model?

We inserted the following clarifying statement in Section 2

- "…with our timestep of 1/8th of a year, computed using Runge-Kutta 4 integration, …"

References:

Alberti, M., Asbjornsen, H., Baker, L. A., Brozovic, N., Drinkwater, L. E., Drzyzga, S. A., et al. (2011). Research on Coupled Human and Natural Systems (CHANS): Approach, Challenges, and Strategies. *Bulletin of the Ecological Society of America*, *92*(2), 218–228. https://doi.org/10.1890/0012-9623-92.2.218

Boyd, D., Hartter, J., Boag, A. E., Jain, M., Stevens, K., Nicholas, K. A., et al. (2017). Top 40 questions in coupled human and natural systems (CHANS) research. *Ecology and Society*, *22*(2).

Breach, P. A., & Simonovic, S. P. (2021). ANEMI3: An updated tool for global change analysis. *PLOS ONE*, *16*(5), e0251489. https://doi.org/10.1371/journal.pone.0251489

Di Vittorio, A. V., Sinha, E., Hao, D., Singh, B., Calvin, K. V., Shippert, T., et al. (2025). E3SM-GCAM: A Synchronously Coupled Human Component in the E3SM Earth System Model Enables Novel Human-Earth Feedback Research. *Journal of Advances in Modeling Earth Systems*, *17*(6), e2024MS004806. https://doi.org/10.1029/2024MS004806

Jiang, H., Simonovic, S. P., & Yu, Z. (2022). ANEMI_Yangtze v1.0: a coupled human – natural systems model for the Yangtze Economic Belt – model description. *Geoscientific Model Development*, *15*, 4503–4528.

Liu, J., Dietz, T., Carpenter, S. R., Alberti, M., Folke, C., Moran, E., et al. (2007). Complexity of Coupled Human and Natural Systems. *Science*, *317*(September), 1513–1517.

Motesharrei, S., Rivas, J., Kalnay, E., Asrar, G. R., Busalacchi, A. J., Cahalan, R. F., et al. (2017). Modeling sustainability: Population, inequality, consumption, and bidirectional coupling of the Earth and human Systems. *National Science Review*, *3*(4), 470–494. https://doi.org/10.1093/nsr/nww081

Sang, S., Li, Y., Zong, S., Yu, L., Wang, S., Liu, Y., et al. (2025). The modeling framework of the coupled human and natural systems in the Yellow River Basin. *Geography and Sustainability*, 100294. https://doi.org/10.1016/j.geosus.2025.100294

Qu, W., Shi, W., Zhang, J., & Liu, T. (2020). T21 China 2050: A Tool for National Sustainable Development Planning. *Geography and Sustainability*, *1*(1), 33–46. https://doi.org/10.1016/j.geosus.2020.03.004

Li, Y., Sang, S., Mote, S., Rivas, J., & Kalnay, E. (2023). Challenges and opportunities for modeling coupled human and natural systems. *National Science Review*, *10*(7), nwad054. https://doi.org/10.1093/nsr/nwad054

Pedercini, M., Arquitt, S., Collste, D., & Herren, H. (2019). Harvesting synergy from sustainable development goal interactions. *Proceedings of the National Academy of Sciences*, *116*(46), 23021–23028. https://doi.org/10.1073/pnas.1817276116

Ye, Q., Liu, Q., Swamy, D., Gao, L., Moallemi, E. A., Rydzak, F., & Eker, S. (2024). FeliX 2.0: An integrated model of climate, economy, environment, and society interactions. *Environmental Modelling & Software*, *179*, 106121. https://doi.org/10.1016/j.envsoft.2024.106121

Yang, S., Dong, W., Chou, J., Feng, J., Yan, X., Wei, Z., et al. (2015). A brief introduction to BNU-HESM1.0 and its earth surface temperature simulations. *Advances in Atmospheric Sciences*, *32*(12), 1683–1688. https://doi.org/10.1007/s00376-015-5050-6

References:

Rajah, J. K., Blanz, B., Kopainsky, B., & Schoenberg, W. (2025). An endogenous modelling framework of dietary behavioural change in the fully coupled human-climate FRIDA v2.1 model. *EGUsphere*, *2025*, 1-35.

---

## Author Response (AR2)

Dear Dalei,

Thank you for the clarification and additional information that we have repeated below.

**The 2rd reviewer stated that**

"I note that the authors have largely rejected my suggestions to improve the manuscript, making only some light editing in response to comments. Personally I am unpersuaded by their response to my first substantive comment - that this paper is premature given underlying papers providing details on component modules are unpublished, making it difficult for reviewers (or readers if published) to evaluate the overall model. This is an editorial call though so I defer to editorial judgement on this and journal policy. I do note that underlying code is available in the zenodo repository, so I guess all code, equations, calibration data etc are available for inspection for an interested reader."

Please carefully consider all the comments from the 2rd reviewer in the first round review and clearly state why the suggestions/comments are rejected.

In the first round of review, Reviewer #1 wrote the following, which appears to be the source of the comment highlighted to us in red:

Firstly, I note that evaluating the model as presented is challenging because only high-level information on model components is presented. Details of functional forms, parameters, and calibration for individual model components are referenced to other papers that are, in most case, still unpublished. Given these components may well change as part of that publication process, and because, until publication, these important details are unavailable for review, it seems it might be appropriate to wait on publication of the full model until those processes are finalized. I believe this is an editorial decision.

We had written in reply to Reviewer #1 in our response letter the following...

We respectfully disagree with the referee on this point. Publishing more detailed papers that outline the entirety of the equations for this entirely new model requires a paper like this. This paper discusses the philosophy and broad approach of FRIDA, as well as its validation, and calibration procedures – all of which those authors need a citation for. As a research team we decided the best way to break this chicken-and-egg problem was to produce a broad overview paper, demonstrating the approach under which FRIDA was developed so that authors publishing on specific portions of FRIDA can have an organizing framework they could write within while not having to repeat large sections of this paper in each of those papers. To produce an entire detailed documentation of FRIDA in a single paper would necessitate a paper of prohibitive length. Although, we have made direct reference to the full source code of the model in the first sentence of Section 2 where we introduce FRIDA.

"FRIDA is a global model that focuses on closing the system-wide feedback loops (processes) that cut across the climate and human systems (see Schoenberg et al., (2025) for model source code)."

We understand how important open access and reproducibility is. To that end we have developed and released the FRIDA model as open source under the MIT license since its inception. The model is hosted here: https://github.com/metno/WorldTransFRIDA on GitHub, and the specific version 2.1 has been hosted on Zenodo here: https://zenodo.org/records/15310860. We have also made available all simulation data here: https://zenodo.org/records/15396799 including all source code necessary to generate that data here: https://github.com/BenjaminBlanz/WorldTransFrida-Uncertainty. We remain firm in our belief that it would require a single paper of prohibitive length and complexity to publish a full listing of all model equations and their documentation. We have the ability to produce an Appendix that contains a table listing every equation, and its units. This table contains over 3000 equations and is hundreds of pages long. Likewise, we view the entirety of Section 3 as documentation for how the model works. If there are specific places where you feel we need to add more detail, or specifics to make the work more broadly understood we would be happy to oblige. In addition, we have changed the title of the paper to make clear that this paper is an overview of the model as was suggested.

An overview of FRIDA v2.1: A feedback-based, fully coupled, global integrated assessment model of climate and humans

All of the key formulations, parameters, and their documentation will be published in this collection that this paper is part of (see for example this published paper: <a href="https://doi.org/10.5194/gmd-18-5997-2025">https://doi.org/10.5194/gmd-18-5997-2025</a>). We still feel strongly that this paper needs to come first for the reasons we stated in our original review response.

Thank you.